# *Efficient value synthesis* in the orbitofrontal cortex explains how loss aversion adapts to the ranges of gain and loss prospects

**Jules Brochard[1,2,3], Jean Daunizeau[1,2,3]***

[1]Sorbonne Université, Paris, France; [2]Institut du Cerveau, Paris, France; [3]INSERM UMR S1127, Paris, France

**Abstract** Is irrational behavior the incidental outcome of biological constraints imposed on neural information processing? In this work, we consider the paradigmatic case of gamble decisions, where gamble values integrate prospective gains and losses. Under the assumption that neurons have a limited firing response range, we show that mitigating the ensuing information loss within artificial neural networks that synthetize value involves a specific form of self-organized plasticity. We demonstrate that the ensuing efficient value synthesis mechanism induces value range adaptation. We also reveal how the ranges of prospective gains and/or losses eventually determine both the behavioral sensitivity to gains and losses and the information content of the network. We test these predictions on two fMRI datasets from the OpenNeuro.org initiative that probe gamble decision-making but differ in terms of the range of gain prospects. First, we show that peoples' loss aversion eventually adapts to the range of gain prospects they are exposed to. Second, we show that the strength with which the orbitofrontal cortex (in particular: Brodmann area 11) encodes gains and expected value also depends upon the range of gain prospects. Third, we show that, when fitted to participant's gambling choices, self-organizing artificial neural networks generalize across gain range contexts and predict the geometry of information content within the orbitofrontal cortex. Our results demonstrate how self-organizing plasticity aiming at mitigating information loss induced by neurons' limited response range may result in value range adaptation, eventually yielding irrational behavior.

**\*For correspondence:**
jean.daunizeau@gmail.com

**Competing interest:** The authors declare that no competing interests exist.

## Editor's evaluation

This valuable manuscript proposes a neural network mechanism for range adaptation for value-based decision making. The authors present solid evidence for the proposed mechanism.

## Introduction

Why do we maintain unrealistic expectations or engage in irresponsible conduct? Maybe one of the most substantial and ubiquitous violations of rationality is peoples' sensitivity to modifications and/or manipulations of contextual factors that are irrelevant to the decision problem (*Kahneman, 2011*; *Seymour and McClure, 2008*). A prominent example is that people's attitude towards risk depends upon whether alternative choice options are framed either in terms of losses or in terms of gains (*Kahneman and Tversky, 2012*). More generally, many forms of irrational behaviors stem from peoples' relative (as opposed to absolute) perception of value, that is: value is perceived in relation to a contextual reference point (*Seymour and McClure, 2008*; *Srivastava and Schrater, 2011*). Because

it provides a mechanistic interpretation of such relative/context-dependent decision processes, range adaptation in value-sensitive neurons is currently under intense scrutiny (*Louie et al., 2013*; *Rangel and Clithero, 2012*; *Rigoli et al., 2016*; *Rustichini et al., 2017*; *Steverson et al., 2019*). Neural range adaptation was first observed in the brain's perceptual system: neurons in the retina normalize their response to incoming light in their receptive field w.r.t. to the illumination context, such that output firing rates span the variation range of surrounding light intensities (see *Carandini and Heeger, 2011* for a review). Importantly, this neural mechanism provides a principled explanation for some forms of context-dependent visual illusions (*May and Zhaoping, 2016*; *Pooresmaeili et al., 2013*; *Troscianko and Osorio, 2023*). The underlying assumption here is that perceptual neurons *transmit* the information that they receive, i.e., a neuron's input is the physical quantity that is signaled to the brain (e.g. light intensity within a certain frequency band), whereas the neuron's output is the percept (e.g. perceived amount of red). In turn, range adaptation (to a neuron's input signal) directly induces perceptual context-dependent effects. But neural range adaptation may not be a glitch in the brain's perceptual system. Rather, it may be understood as the brain's best attempt to produce optimal information processing, given its own hard-wired biological constraints. This is the perspective afforded by *efficient coding*: light-sensitive neurons *should* adapt their firing properties to mitigate visual information loss resulting from their limited firing range (*Brenner et al., 2000*; *Laughlin, 1981*; *Valerio and Navarro, 2003*; *Wark et al., 2007*). In other words, range adaptation would improve the average reliability of neural information transmission, at the cost of inducing visual illusions in some circumstances.

Range adaptation was later evidenced on neural value coding, i.e., value-sensitive neurons was shown to normalize their response w.r.t. the set of alternative options within a given choice context and/or to the recent history of experienced/prospective rewards (*Louie et al., 2013*; *Rangel and Clithero, 2012*). In particular, gradual range adaptation effects have been the focus of intense research over the past decade, because they hold the promise of explaining persistent forms of irrational behavior. In line with the existing literature on value processing in the brain, they have been repeatedly documented in non-human primates, mostly using electrophysiological recordings in the orbitofrontal cortex or OFC (*Conen and Padoa-Schioppa, 2019*; *Kobayashi et al., 2010*; *Padoa-Schioppa, 2009*; *Tremblay and Schultz, 1999*; *Yamada et al., 2018*), though similar effects have been demonstrated in the anterior cingulate cortex (*Cai and Padoa-Schioppa, 2012*) and the amygdala (*Bermudez and Schultz, 2010*; *Saez et al., 2017*). Although comparatively sparser, neural evidence for gradual value normalization in the human OFC and ventral striatum also exists (*Burke et al., 2016*; *Cox and Kable, 2014*; *Elliott et al., 2008*). Importantly, when included in computational models of value-based decision-making, efficient coding in value-sensitive neurons partially explains specific forms of irrational behavior away (*Polanía et al., 2019*; *Zimmermann et al., 2018*).

Having said this, the neurophysiological bases of range adaptation in value-sensitive neurons are virtually unknown and their behavioral consequences are debated (*Khaw et al., 2017*; *Rustichini et al., 2017*). For example, that overt preferences do not shift along with the observed changes in the value-sensitivity of OFC neurons is puzzling. A possibility is that range adaptation in OFC neurons may be '"undone' by Hebbian-like plasticity mechanisms that fine-tune the synaptic efficacy of downstream 'value comparison' neurons (*Padoa-Schioppa and Rustichini, 2014*; *Rustichini et al., 2017*). Implicit in this reasoning is the assumption that option values are typically considered as input signals to value-sensitive OFC neurons, which then transmit this information to downstream decision systems, in analogy to the transmission of light-intensity information by neurons in the retina (*Louie and Glimcher, 2012*). But another possibility is that value coding in OFC neurons departs from the logic of efficient information transmission in the visual system (*Burke et al., 2016*; *Conen and Padoa-Schioppa, 2019*). For example, OFC neurons may be constructing (as opposed to receiving) value signals, out of input signals conveying information about possibly conflicting decision-relevant attributes (*Lim et al., 2013*; *O'Doherty et al., 2021*; *Pessiglione and Daunizeau, 2021*; *Raghuraman and Padoa-Schioppa, 2014*). We refer to this as *value synthesis*. In what follows, we consider the paradigmatic case of risky decisions, which require integrating attributes such as prospective gains and losses. Here, value synthesis implies weighing prospective gains and losses, such that the ensuing subjective value effectively arbitrates between pro- versus anti-gamble behavioral tendencies. Our working assumption is twofold: (i) attribute-integration units in the OFC receive idiosyncratic mixtures of signals from attribute-specific units, and (ii) value is read out from integration units using a dedicated population code. This enables us to extend existing models of efficient coding to the case of

value synthesis. Under mild conditions regarding units' response properties, we show that a simple form of self-organized plasticity between attribute-specific and attribute-integration units can mitigate information loss induced by the limited firing range of attribute-integration units. Under such an *efficient value synthesis* scenario, OFC neurons would not adapt to the range of (output) values; rather, they would adapt to the range of their native input signals. Importantly, the ensuing neural and behavioral consequences depend upon how the underlying self-organized plasticity mechanism modifies the shape of integration units' receptive fields over the spanned gain/loss domain. In this work, we derive the self-organized plasticity rule that operates efficient value synthesis, highlight its neural and behavioral consequences, and test the ensuing quantitative predictions against behavioral and neural data.

In particular, we show how the ranges of value-relevant attributes (i.e. here: prospective gains and losses) eventually determine the geometry of information encoded in the population of integration units, as well as the landscape of output value signals over the spanned domain of attributes. This is important, because the latter drives peoples' loss aversion, i.e., their tendency to overweigh prospective losses over prospective gains when considering whether to accept or reject risky gambles. We then perform an entirely novel re-analysis of two independent fMRI datasets, which are made available in the context of the https://openneuro.org/ initiative (*Poldrack et al., 2013*). In both studies, participants are asked to accept or reject a series of gambles, but the two studies differ w.r.t. to the range of prospective gains. First, we test the neural and behavioral predictions of efficient value synthesis. In particular, we provide evidence that peoples' loss aversion progressively adapts to the range of prospective gains. We also evaluate the neural predictions of the efficient value synthesis scenario by quantifying the geometry of information in five subregions of the OFC. Finally, we fit the artificial neural network models to peoples' gambling choices and show that, only when endowed with the self-organized plasticity mechanism for efficient value synthesis do they generalize across gain range context (i.e. across participants' groups) and predict (out-of-sample) the full geometry of information content within the OFC.

## Results

### Efficient value synthesis: Computational mechanism and model predictions

We consider that value synthesis is operated by neural networks composed of two layers (see Methods): (i) an attribute-specific layer further divided into two sets of units that differ in terms of their inputs (either trial-by-trial gains or losses), and (ii) an integration layer receiving outputs from both attribute-specific units (see *Figure 1A* below).

By assumption, the gamble's value $V_t$ at trial $t$ is read out from the response pattern in the integration layer using a linear population code (*Pessiglione and Daunizeau, 2021*):

$$\begin{cases} V_t = \sum_k w^{(k)} z_t^{(k)} \\ z_t^{(k)} = f_z^{(k)}\left(v_t^{(k)}\right) \\ v_t^{(k)} = \sum_{i,j} C^{(i,j,k)} x_t^{(i,j)} \\ x_t^{(i,j)} = f_x^{(j)}\left(u_t^{(i)}\right) \end{cases} \tag{1}$$

where $z_t^{(k)}$ (respectively, $x_t^{(i,j)}$) is the output response of the $k^{th}$ integration unit (respectively, $j^{th}$ unit within the $i^{th}$ attribute sublayer), $w^{(k)}$ is the corresponding value readout weight in the population code, $f_z^{(k)}$ is its input-output activation function, $v_t^{(k)}$ is the input signal to the $k^{th}$ integration unit (which is made of a weighted mixture of attribute units' outputs), $C^{(i,j,k)}$ is its connection strength with the $j^{th}$ unit of the $i^{th}$ attribute sublayer, and $u_t^{(i)}$ is the input signal to the $i^{th}$ attribute sublayer (i.e. either gain or loss information) at trial $t$. Importantly, we assume that units have a bounded response range, i.e., their firing rate cannot exceed some predefined physiological limit. Recall that

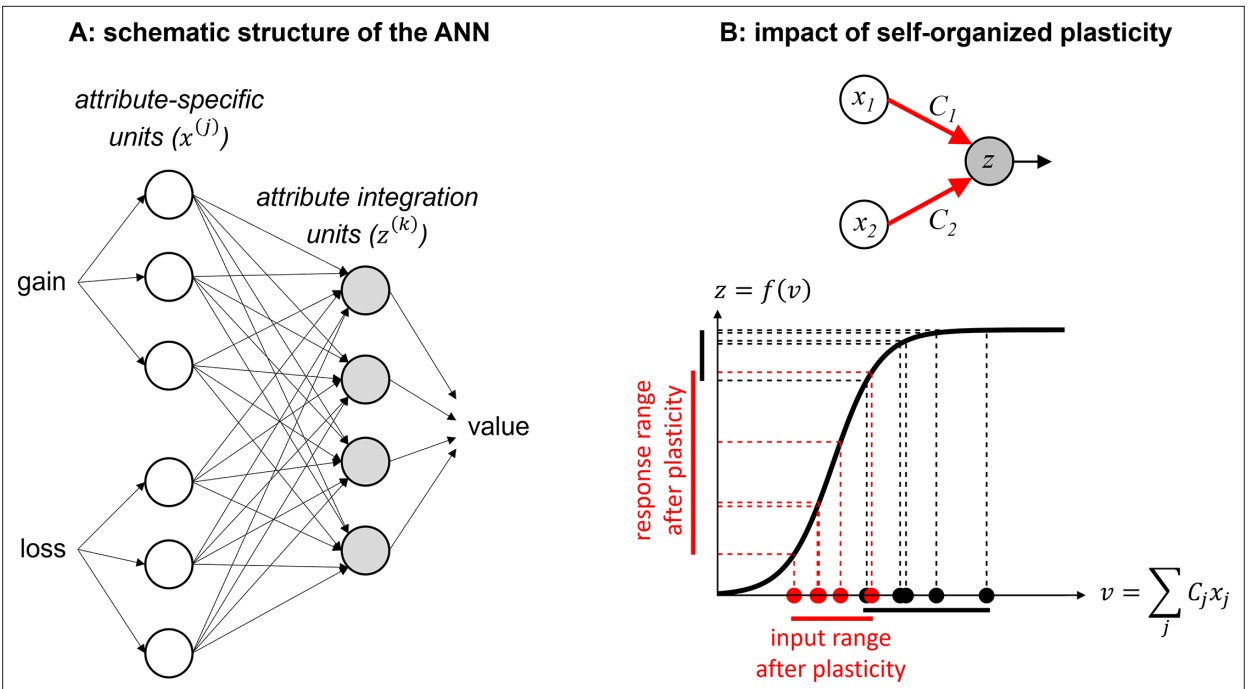

**Figure 1.** Efficient value synthesis. (**A**) Schematic structure of the artificial neural network (ANN). Trial-by-trial prospective gains (G) and losses (L) first enter attribute-specific units (white circles), which then send their outputs to integration units (gray circles). The outputs of these units are then combined to yield trial-by-trial gamble values, using a linear population code. (**B**) The impact of self-organized plasticity on integration units' response. Integration units receive a weighted mixture (v) of attribute units' outputs (x), and their firing response is a sigmoidal mapping $z=f(v)$ of their input. In principle, inputs to integration units (black dots) may sample the saturating range of their sigmoidal activation function (black vertical bar on the y-axis). However, self-organized plasticity modifies the connection strengths between attribute and integration units, such that the ensuing inputs to integration units (red dots) eventually fall within the responsive range of the activation function (red vertical bar on the y-axis).

The online version of this article includes the following figure supplement(s) for figure 1:

**Figure supplement 1.** Summary of sigmoidal integration units' response profiles.

**Figure supplement 2.** Efficiency of value synthesis.

**Figure supplement 3.** Summary of integration units' response profiles after self-organized plasticity.

**Figure supplement 4.** Analysis of the information content within the artificial neural networks (ANN's) integration layer.

the main mechanistic constraint that acts on a neuron's firing rate is its action potential's refractory period, which depends upon how long it takes ion channels to complete a whole voltage activate-deactivate cycle (about a few milliseconds). Typically, even fast-spiking neocortical neurons cannot fire at a frequency higher than about 500 or 600 Hz (*Wang et al., 2016*). In our context, this can be simply modeled using saturating (more precisely: sigmoidal) input-output activation functions. In turn, the receptive field of integration units over the bidimensional domain of prospective gains and losses can exhibit arbitrary idiosyncratic shapes (see e.g. *Figure 1—figure supplement 1*).

In principle, artificial neural networks (ANNs) described by *Equation 1* can be trained to output almost any subjective value landscape over the bidimensional domain spanned by prospective gains and losses. In particular, they can be trained to output actions' expected value (EV), which would yield a neural implementation of normative decision-making. For example, consider a gamble decision that entails a 50/50 chance of either earning an amount G of money or losing an amount L. In this case, the gamble's EV is simply the sum of prospective gains and losses, weighted by their occurrence probability, i.e., $EV = \frac{1}{2}G - \frac{1}{2}L$. Training the network to operate this kind of (linear) value synthesis is trivial. If the units' responses were perfectly reliable (i.e. noiseless), then this would eventually yield rational behavioral responses. But, due to the limited response range of neural units, even small amounts of noise on the integration layer may induce a strong loss of information on value. This happens when inputs to integration units fall outside their responsive range, because it saturates the output contrast. By analogy with efficient coding in the brain's perceptual systems, *efficient value synthesis* would rely

upon some form of unsupervised adaptation mechanism to mitigate such information loss. For ANNs that obey *Equation 1*, one can show that the following self-organized plasticity rule operates efficient value synthesis (see Methods section):

$$\Delta C_t^{(i,j,k)} = \alpha \left(1 - \beta\right) \Delta C_{t-1}^{(i,j,k)} + \alpha \frac{\beta}{\sigma_z^{(k)}} \left(1 - 2z_t^{(k)}\right) x_t^{(i,j)} \tag{2}$$

where $\Delta C_t^{(i,j,k)}$ is the trial-by-trial change in connection strength between attribute and integration units and $\sigma_z^{(k)}$ is the slope parameter of the $k^{\text{th}}$ integration unit's activation function. Here, $\alpha$ and $\beta$ determine both the magnitude and temporal scale at which self-organized plasticity unfolds (see Methods section). *Equation 2* modifies the network connections such that inputs to integration units (i.e. $v_t^{(k)}$ in *Equation 1*) fall within the responsive range of their activation function, which maximizes the output variability induced by attribute inputs (see *Figure 1B*). This eventually improves the behavioral resilience of the system to neural noise (see *Figure 1—figure supplement 2*). The plasticity rule in *Equation 2* is 'self-organized' in the sense that it does not require any teaching or feedback signal. It is also 'local,' in that a connection only changes in response to (the recent history of) the outputs of the corresponding pair of connected units. Finally, it is 'anti-Hebbian,' in that connections tend to weaken when units co-activate. The exact form of self-organized plasticity that operates efficient

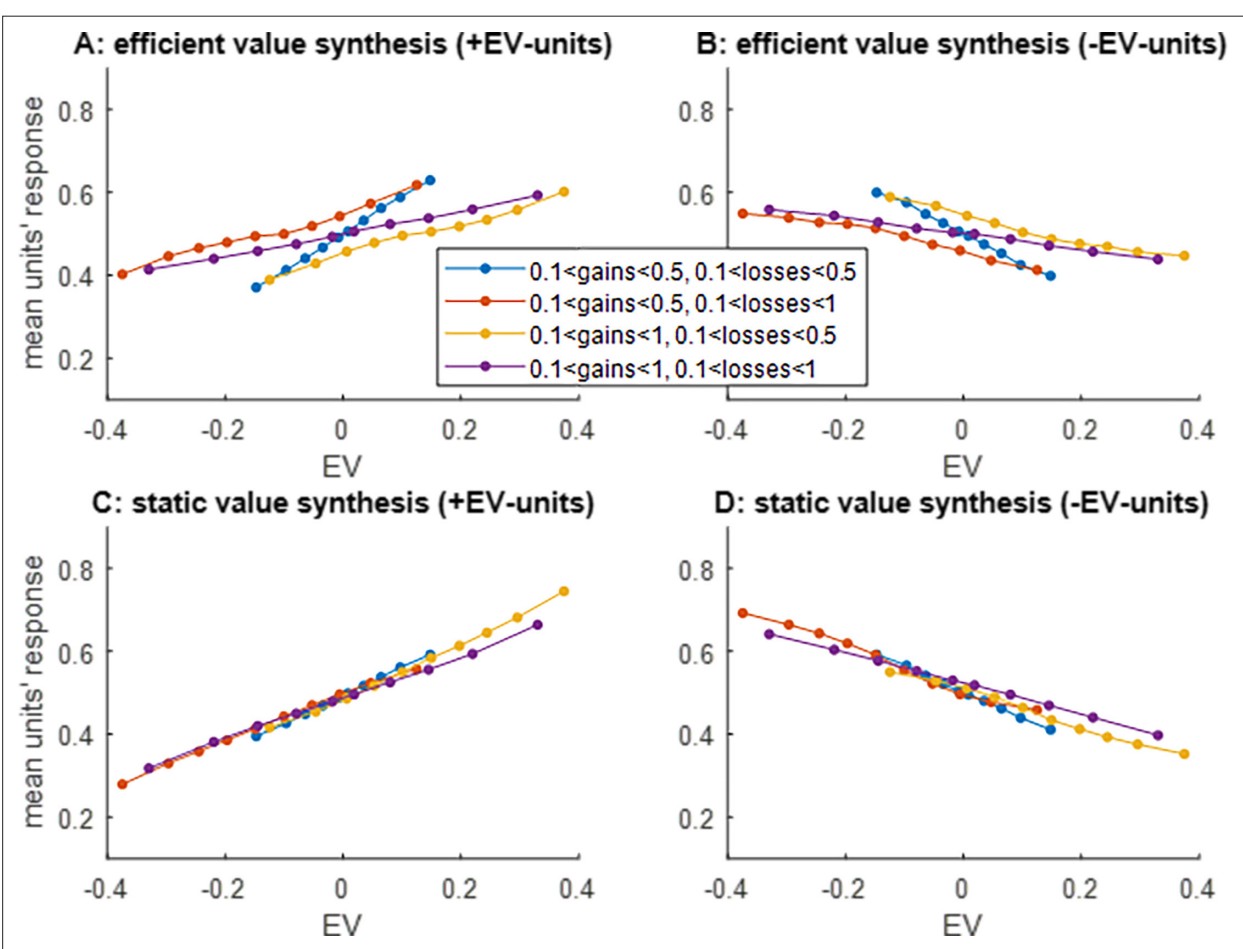

**Figure 2.** Apparent value range adaptation. (**A**) The average units' response (y-axis) to pairs of prospective gains and losses that fall within predefined expected value (EV) bins (x-axis) is shown, while artificial neural networks (ANNs) that operate efficient value synthesis are exposed to four different spanned gain/loss domains (blue: narrow ranges of gains and losses, violet: wide ranges of gains and losses, red: narrow gain range and wide loss range, yellow: wide gain range and narrow loss range, see main text). Only integration units that correlate positively with EV are shown. (**B**) integration units that correlate negatively with EV, same format as panel (**A**). C/D: same format as panels (**A** and **B**), for ANNs that operate static value synthesis.

value synthesis depends upon the units' activation function, but these properties generalize to any nonlinear activation function that is continuous and monotonically increasing.

Setting $\alpha = 0$ or $\beta = 0$ yields non-adaptive value synthesis, whereby the value readout is independent of the history of spanned prospective gains and losses. In what follows, we refer to this as *static* value synthesis. Otherwise (i.e. when $\alpha > 0$ and $\beta > 0$), the self-organized plasticity mechanism in *Equation 2* progressively modifies the shape of integration units' receptive fields over the spanned gain/loss domain (see *Figure 1—figure supplement 3*). This has three main notable consequences.

First, this eventually induces apparent value range adaptation, i.e., the average response of integration units settles between two (almost) invariant bounds which systematically map onto the gambles' EV extrema over the spanned gain/loss domain. In addition, despite potentially strong nonlinearities in the receptive fields of integration units, the average relationship between their activity and gambles' EV is almost linear. Interestingly, this apparent value range adaptation effect comes in two variants, depending upon the sign of the correlation between units' activity and gambles' EV.

*Figure 2* exemplifies the apparent value range adaptation effect. This is an average of over 1000 Monte-Carlo simulations, where we randomized the trained connections of ANNs that operate (either *efficient* or *static*) value synthesis prior to exposing them to four different series of 256 decision trials made of prospective gains and losses with a predefined range. More precisely, we considered two ranges (either narrow or wide) for both prospective gains and losses, and exposed the ANNs to each of the 2×2 range combinations. By chance, some units become less sensitive to prospective gains than to prospective losses: those will tend to show a negative correlation with EV ('-EV-units'). Interestingly, the responses of '+EV-units' and '-EV-units' shown on panels A and B are reminiscent of value range adaptation effects evidenced using electrophysiological recordings of OFC neuron activity (*Conen and Padoa-Schioppa, 2019*; *Padoa-Schioppa, 2009*). In particular, one can see that the slope of the relationship between EV and integration units' activity only depends upon the EV range, and not upon the actual bounds of the spanned domain of EVs. However, this value range adaptation effect is only apparent, in the sense that integration units do not respond to value: they respond to prospective gains and losses. This is important, because the underlying plasticity mechanism reacts to the relative range of spanned gains and losses, which is partially orthogonal to the induced range of EVs. On thus needs to consider the shape of the spanned domain of decision-relevant attributes to properly understand the neural and behavioral consequences of efficient value synthesis.

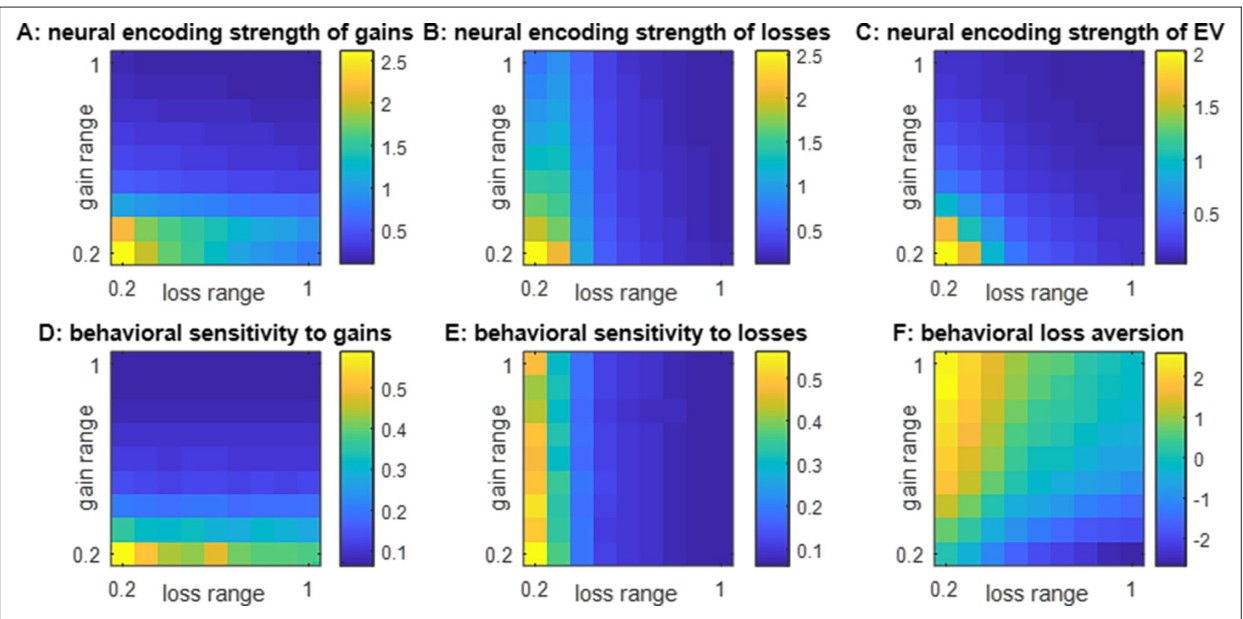

**Figure 3.** Impact of spanned ranges of gains and losses. (**A**) The neural encoding strength of gains (color code: from blue -minimal encoding strength- to yellow -maximal encoding strength-) is shown as a function of the spanned range of losses (x-axis, range increases from left to right) and gains (y-axis, range increases from bottom to top). Note that the maximal range of prospective gains and losses is arbitrarily set to unity. (**B**) Neural encoding strength of losses, same format. (**C**) Neural encoding strength of EV, same format. (**D**) Behavioral sensitivity to gains, same format. (**E**) Behavioral sensitivity to losses, same format. (**F**) Behavioral loss aversion, same format.

Second, self-organized plasticity changes the information content within the integration layer. To show this, we measure the neural dissimilarity between response patterns within the integration layer for any pair of decision trials, and then quantify its change when pairwise differences in either prospective gains, losses or EV vary. We define the 'neural encoding strength' of gains, losses, or EV in terms of the gradient of neural dissimilarity per unit of absolute difference in gains, losses, or EV, respectively (see *Figure 1—figure supplement 4*). In brief, the neural encoding strength of gains (resp. losses) decreases when the spanned range of gains (resp. losses) increases. This is also true for EV, whose neural encoding strength in the integration layer decreases when either the range of gains or losses increases.

Third, this modifies the relative sensitivity of readout value to prospective gains and losses. We quantify the sensitivity of readout value w.r.t. to its constituent attributes in terms of its gradient per unit of prospective gain and losses. In brief, the sensitivity of the ANN's readout value signal to prospective gains (resp. losses) decreases when the spanned range of gains (resp. losses) increases. In turn, behavioral loss aversion - as defined by the ratio between loss and gain sensitivities- increases (resp. decreases) when the range of gains (resp. losses) increases. This is important, because this suggests how peoples' behavior will deviate from classical decision theory and exhibit irrational context-dependency effects.

*Figure 3* below summarizes the impact of the shape of the spanned bidimensional gain/loss domain. This is an average of over 1000 Monte-Carlo simulations, where the trained ANN's connections are randomized prior to self-organizing according to *Equation 2* in response to a series of 256 decision trials made of prospective gains and losses with a predefined range, which is systematically varied.

Strikingly, the behavior will exhibit positive (respectively, negative) loss aversion when the context is favorable (respectively, unfavorable), i.e., when the spanned range of gains is greater (respectively, smaller) than that of losses (*Figure 3F*). Also, behavioral loss aversion is expected to be neutral when the spanned ranges of gains and losses are comparable (symmetrical gain/loss domain).

In addition to the main effects described above, one can see that there is a cross-attribute spillover effect, such that the behavioral and neural sensitivities to prospective gains (respectively, losses) also decrease when the spanned range of losses (respectively, gains) increases (*Figure 3A, B, D and E*). The magnitude of these spillover effects is comparatively weaker and may thus be more difficult to detect in an empirical setting. In fact, the only quantity that is similarly impacted by the ranges of gains and losses is the neural encoding strength of EV (*Figure 3C*). This effect is partly driven by changes in the neural sensitivity to gains and losses (*Figure 3A and B*), which constrains the availability of information on EV within the integration layer. But it also derives from the distortion of the readout value profile (*Figure 3D and E*), which weakens the statistical relationship between EV and integration units' activity patterns. Both are consequences of the changes in integration units' receptive fields that are induced by self-organized plasticity (see *Figure 1—figure supplements 1 and 3*). This eventually translates into an apparent value range adaptation phenomenon that generalizes the univariate effect reported in *Figure 2*.

Note that all these range adaptation effects actually unfold over time, as the network progressively self-organizes in response to prospective gains and losses. Importantly, however, numerical simulations show that the ensuing dynamics of neural and behavioral sensitivities converge, i.e., they eventually reach a steady-state. The convergence rate is governed by the parameter $\beta$, whereas the overall magnitude of these context-dependency effects is determined by the parameter $\alpha$. Although the exact setting of the plasticity magnitude and rate parameters do modify the global magnitude of neural and behavioral sensitivity changes, the results shown in *Figure 3* are representative of the impact of the spanned gain/loss domain's shape.

## Model-free analysis of the NARPS dataset

We now present our re-analysis of the NARPS dataset (*Botvinik-Nezer et al., 2020b*). This dataset includes two studies, each of which is composed of a group of 54 participants who make a series of risky decisions. On each trial, a gamble was presented, entailing a 50/50 chance of gaining an amount G of money or losing an amount L. As in *Tom et al., 2007*, participants were asked to evaluate whether or not they would like to accept or reject the gambles presented to them. In the first study (hereafter referred to as the 'narrow range' group), participants decided on gambles made of gain

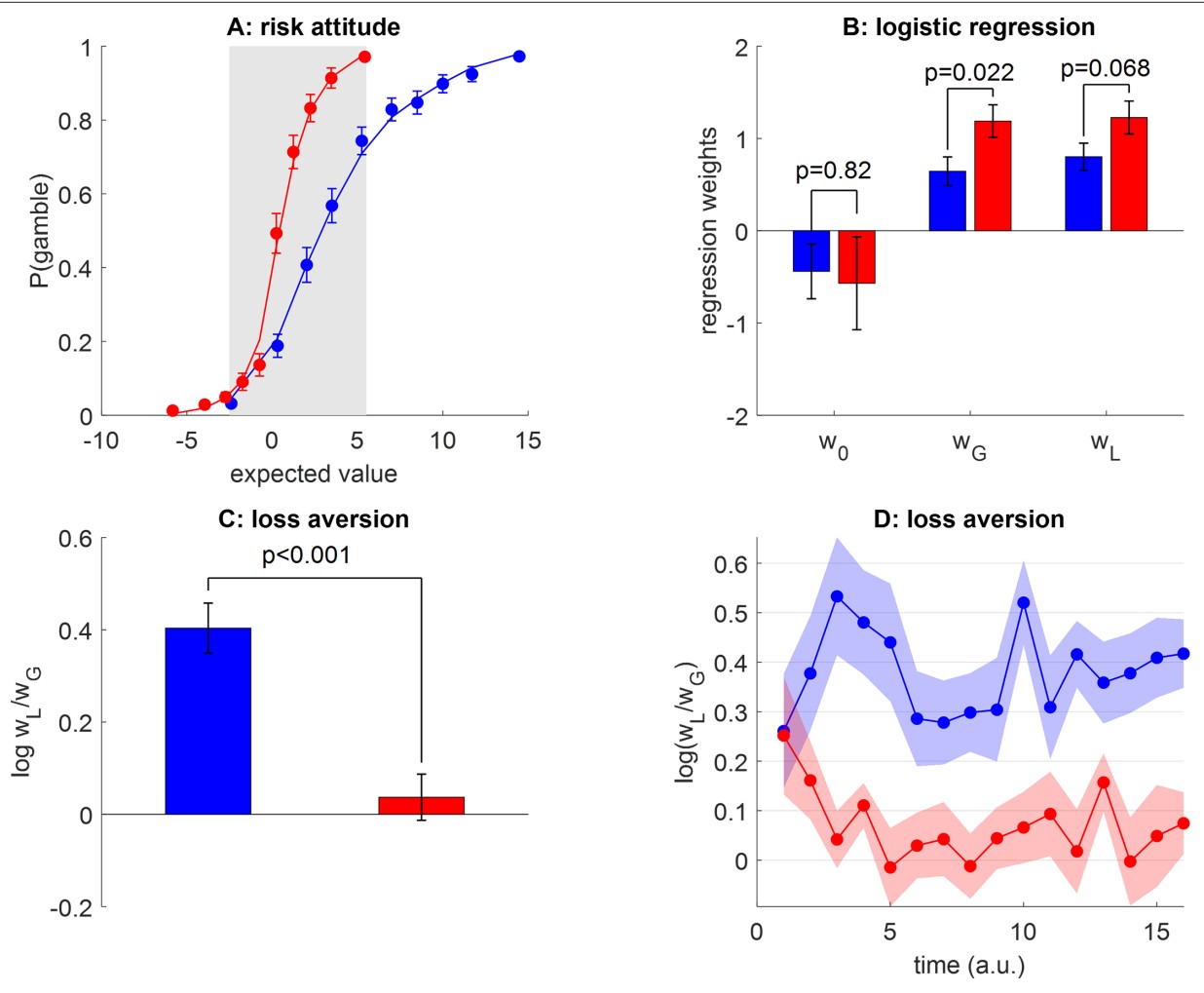

**Figure 4.** Do peoples' loss aversion exhibit range adaptation? (**A**) The group-average probability of gamble acceptance (y-axis) is plotted against deciles of gambles' expected value (EV=(G–L)/2, x-axis), in both groups (red: narrow range, blue: wide range). Dots show raw data (error bars depict s.e.m.), and plain lines show predicted data under a logistic regression model (see main text). The gray-shaded area highlights the range of expected values that is common to both groups. (**B**) Estimates of gamble bias ($w_0$) as well as sensitivity to gains ($w_G$) and losses ($w_L$) for both groups, under the logistic model (same color code as panel A, errorbars depict s.e.m.). (**C**) Average loss aversion (y-axis) is plotted for both groups (same color code, errorbars depict s.e.m.). (**D**) Temporal dynamics of group-average loss aversion (log $w_L/w_G$, y-axis, same color code) are plotted against time (a.u., x-axis). Shaded areas depict s.e.m.

The online version of this article includes the following figure supplement(s) for figure 4:

**Figure supplement 1.** Postdiction error and out-of-sample predictions of the logistic model.

and loss levels that were sampled from the same range (G and L independently varied between 5\$ and 20\$). In the second study (hereafter: the 'wide range' group), gain levels scaled to double the loss levels (L varied between 5\$ and 20\$, and G independently varied between 10\$ and 40\$). Importantly, both groups experience the exact same range of losses. In both studies, all 256 possible combinations of gains and losses were presented across trials (see Methods section). Importantly, the gambles' outcomes were not revealed until the end of the experiment.

To begin with, we ask whether peoples' loss aversion exhibits range adaptation, as predicted by efficient value synthesis. In our context, this implies that (i) peoples' gambling rate should depend upon the gain range context (even within the EV range common to both groups), (ii) peoples' behavioral sensitivity to gains should be higher in the narrow gain range group than in the wide gain range group, (iii) within-group averages of loss aversion should be initially similar and then progressively diverge as time unfolds, and (iv) participants from the wide gain range group should eventually exhibit strong loss aversion while participants from the narrow gain range group should be loss-neutral (cf.

symmetrical gain/loss domain). *Figure 4* below summarizes the results of behavioral data analyses that aim at testing these predictions.

Overall, the average gambling rate of people from the wide range group (65±2%) is much higher than that of people from the narrow range group (44±2%), and the group difference is significant (p<10⁻⁴, *F*=44.8, dof=101). This is of course expected, given that people from the wide range group are exposed to gambles with higher value on average. However, and most importantly, within the range of EV that is common to both groups, people from the wide range group are *less* likely to gamble than people from the narrow range group (*Figure 4A*). Here, the average gambling rate of people from the wide range group is 41±3%, whereas it is 54±3% for people in the narrow range group, and this group difference is significant (p=0.003, *F*=9.2). This difference is likely due to the context in which people made these decisions, which is more favorable (higher gain prospects on average) in the wide-range group. This is the hallmark of context-dependency effects.

From *Figure 4A*, it seems that the variation in peoples' tendency to accept risky gambles approximately spans the range of gambles' values that they are exposed to. Under the efficient value synthesis scenario, this apparent value range adaptation effect is due to context-dependent changes in loss aversion. To investigate this effect, we first performed a within-subject logistic regression of trial-by-trial choice data onto gains and losses (including an intercept, see Methods section). In terms of balanced accuracy, this regression accurately explains 91±0.7% (respectively, 87±0.8%) of individual choices in the narrow (respectively, wide) range group (cf. plain lines in *Figure 4A*). A random effect analysis on regression weight estimates shows that all regression weights are significant at the group level (all p<10⁻³), except for the intercept parameters (narrow range: p=0.26, wide range: p=0.14). This implies that peoples' gambling behavior exhibits no systematic bias above and beyond the effects of prospective gains and losses. Regarding group differences, this analysis also failed to identify a group difference in the constant gambling bias ($w_0$: p=0.82, *F*=0.05). However, peoples' sensitivity to gains is significantly higher in the narrow range group than in the wide range group ($w_G$: p=0.022, *F*=17.5), and this difference is almost significant for loss sensitivity ($w_L$: p=0.068, *F*=3.4). This means that increasing the range of gain prospects decreases peoples' sensitivity to gains (and maybe to losses as well, though to a lesser extent; see *Figure 4B*).

We then derived indices of individual loss aversion, which we define as the log-transformed ratio of loss sensitivity to gain sensitivity, i.e., $\log(w_L/w_G)$ (*Tom et al., 2007*). This definition is not confounded by possible behavioral temperature differences between groups of participants. Mean loss aversion indices are shown in *Figure 4C*. We found that people from the wide range group exhibit significant loss aversion (mean loss aversion index=0.41, sem=0.05, p<10⁻⁴) whereas people from the narrow range group do not (mean loss aversion index=0.037, sem = 0.05, p=0.46), and the ensuing group difference is significant (p=0.0031, *F*=9.2). Importantly, inter-individual differences in loss aversion explain the observed inter-individual differences in peoples' gambling rate within the common EV range across all participants (p<10⁻⁴, *F*=39.9, see *Figure 4—figure supplement 1*).

But is this loss aversion difference due to inter-individual trait differences, or did it grow over time as people are exposed to more gambles? To address this question, we repeated the within-subject logistic regression, this time on consecutive chunks of 16 trials (see Methods section). The resulting temporal dynamics of loss aversion are shown in *Figure 3D*. We found no significant time-by-group interaction (p=0.43, *F*=0.61), which is why we report separate (instantaneous) group comparisons. At the start of the experiment (first 16 trials), loss aversion is significant in both groups (wide range: mean loss aversion index=0.26, sem = 0.1, p=0.027, narrow range: mean loss aversion=0.25, sem=0.1, p=0.039), and there is no significant difference between groups (p=0.96, *F*=0.003). However, as time unfolds, loss aversion in both groups tends to spread apart: the difference between groups starts becoming significant after 32 trials (p=0.005, *F*=13.0) and stays significant thereafter (all p<0.05) except for two chunks of trials (p=0.12 and p=0.054). At the end of the experiment (last 16 trials), loss aversion is significant in participants of the wide range group (mean loss aversion=0.41, sem=0.07, p<10⁻⁴) but not in the narrow range group (mean loss aversion=0.07, sem=0.06, p=0.24), and the group difference is significant (p=0.00042, *F*=13.3).

Those results validate the behavioral predictions of the efficient value synthesis scenario. We now wish to test its neural predictions, namely: (i) EV, as well as prospective gains and losses, should be encoded in neural activity patterns within the OFC, (ii) the neural encoding strength of prospective gains should be higher in the narrow gain range group than in the wide gain range group, (iii) the

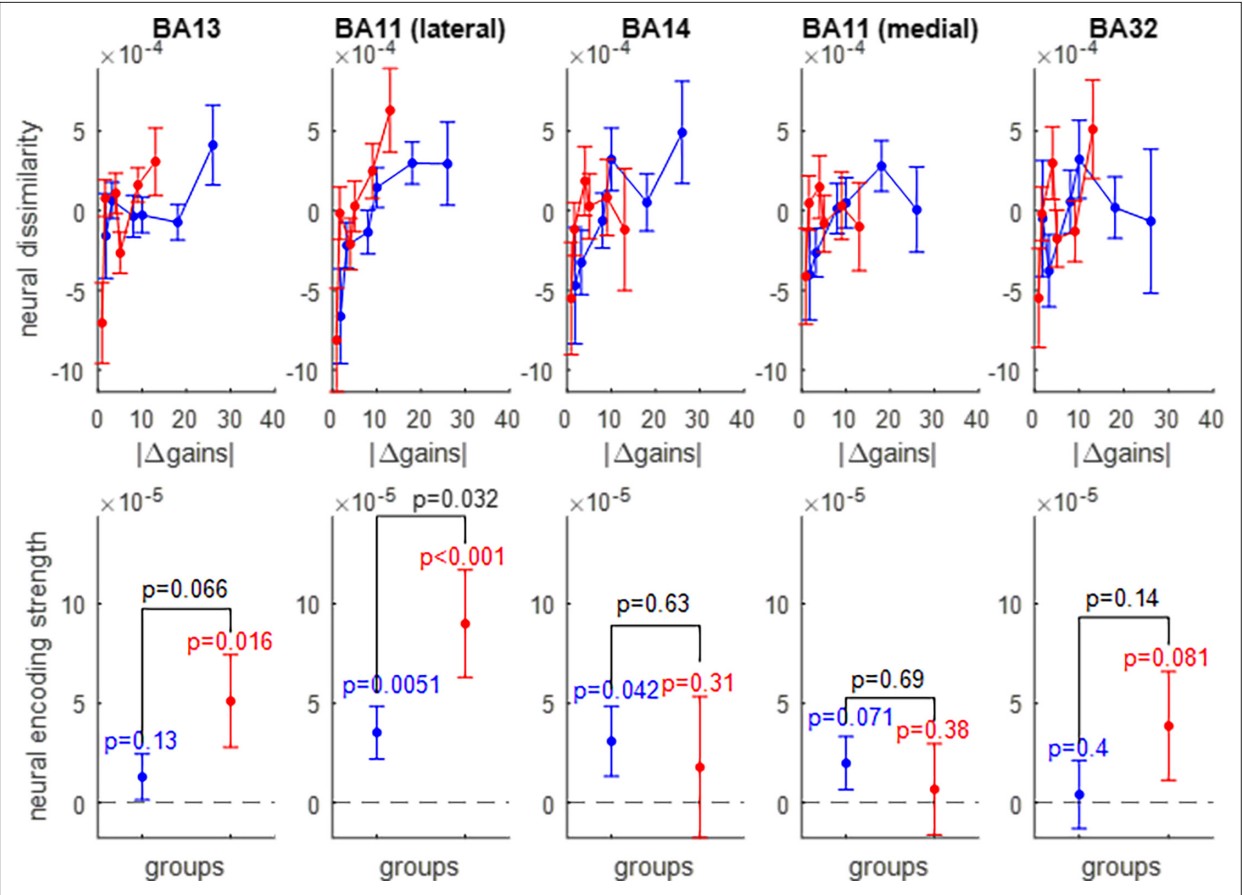

**Figure 5.** Neural encoding of prospective gains. The upper panels show the trial-by-trial neural dissimilarity (y-axis) plotted as a function of trial-by-trial absolute difference in prospective gains (x-axis), for both groups of participants (red: narrow range, blue: wide range), and within each subregion of the OFC. The lower panels show the ensuing neural encoding strength of gains (y-axis), for both groups (same color code), and within each subregion of the OFC. The dotted line indicates the y-axis origin. Errorbars depict s.e.m., and p-values are uncorrected for multiple comparisons.

The online version of this article includes the following figure supplement(s) for figure 5:

**Figure supplement 1.** Gain-representational dissimilarity matrices (RDMs).

neural encoding strength of prospective losses should be equivalent in both groups (up to cross-attribute spillover effects), and (iv) the neural encoding strength of EV should be higher in the narrow gain range group than in the wide gain range group.

We thus extracted the multivariate trial-by-trial BOLD response in five OFC subregions: the lateral and medial parts of Brodmann area 11, Brodmann area 13, Brodmann area 14, and Brodmann area 32 (see Figure 11 in the Methods section). After correcting for between-session and temporal autocorrelation confounding effects (see Methods), we derived the ROI-specific representational dissimilarity matrices and measured the neural encoding strengths of gains (*Figure 5*), losses (*Figure 6*), and EV (*Figure 7*). The corresponding RDMs are shown in *Figure 5—figure supplement 1*, *Figure 6—figure supplement 1,* and *Figure 7—figure supplement 1*, respectively. For the sake of completeness, the results of standard univariate fMRI data analyses can also be eyeballed in *Figure 7—figure supplement 2*.

Of course, prospective gain distances for the wide range-group extend beyond those of the narrow-range group. One can see that, in all subregions of the OFC, neural dissimilarity tends to increase when the absolute difference in prospective gains increases, though this gradient typically attenuates for extreme gain differences. The lateral part of Brodmann area 11 is the only OFC subregion that exhibits a significant encoding of prospective gains in both groups of participants (wide range: p0.0051, narrow range: $p < 10^{-3}$), as well as a significantly higher encoding strength of gains in the narrow range group than in the wide range group (p=0.032).

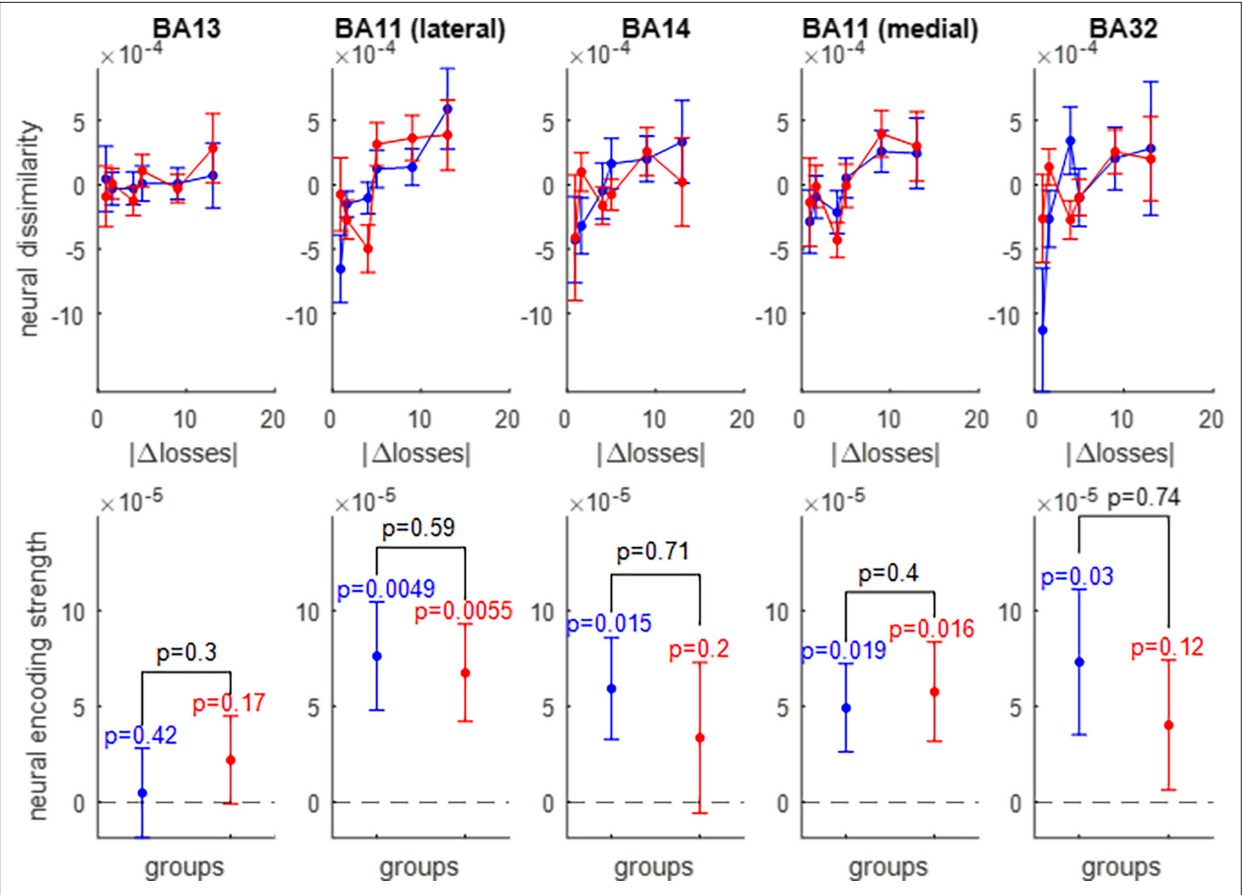

**Figure 6.** Neural encoding of prospective losses. Same format as *Figure 5*.

The online version of this article includes the following figure supplement(s) for figure 6:

**Figure supplement 1.** Loss-representational dissimilarity matrices (RDMs).

Overall, it seems that prospective losses are encoded less strongly in OFC neural activity than prospective gains (though typically about four times stronger in magnitude). Nevertheless, the lateral part of Brodmann area 11 still exhibits a significant encoding of prospective losses in both groups of participants wide range: p=0.0049, narrow range: p=0.0056, without a significant group difference (p=0.59).

The pattern of neural encoding of EV is globally similar to that of prospective gains. Importantly, the lateral part of Brodmann area 11 is the only OFC subregion that exhibits a significant encoding of EV in both groups of participants (wide range: p=0.0061, narrow range: $p<10^{-3}$), as well as a significantly higher encoding strength of EV in the narrow range group than in the wide range group (p=0.0012).

Note that ignoring the fMRI confounding effects does not alter qualitatively the results, though it tends to bury the signal within structured noise, which dampens statistical significance.

In brief, qualitative predictions of the efficient value synthesis scenario at both the behavioral and neural levels have been confirmed (at least in the lateral part of Brodmann area 11). We will now provide further quantitative evidence that efficient value synthesis in the OFC can explain range adaptation of loss aversion.

## Model-based analysis of the NARPS dataset

As can be seen from *Equations 1; 2*, quantitative predictions from the efficient value synthesis scenario actually depend upon model parameters that may vary across individuals. For example, differences in the connectivity matrix $C^{(i,j,k)}$ (at the start of the experiment) and/or value readout weights $w^{(k)}$ can, in principle, account for a broad range of inter-individual differences in gambling behavior (and, possibly, in the neural encoding strength of prospective losses and gains). This raises the question:

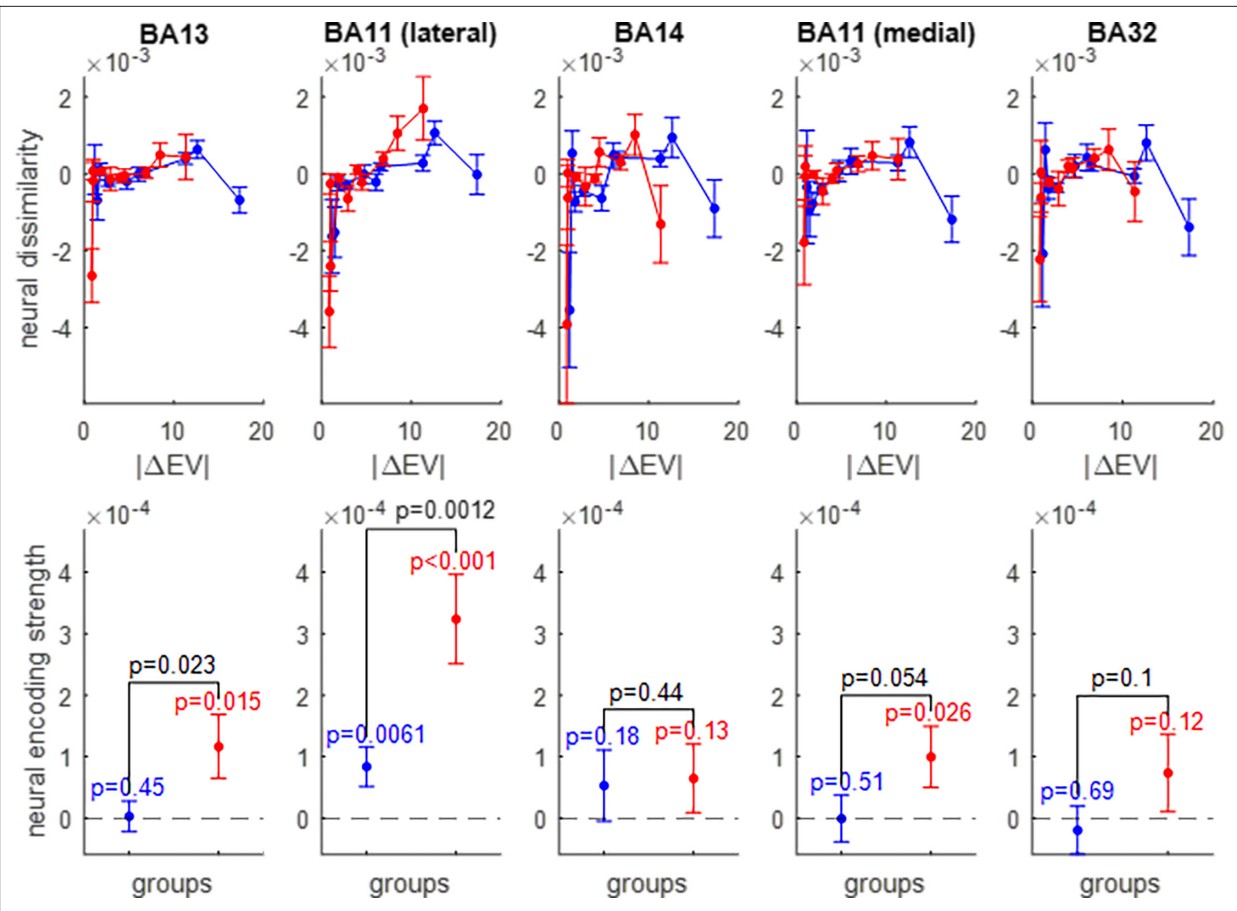

**Figure 7.** Neural encoding of expected value (EV). Same format as *Figure 5*.

The online version of this article includes the following figure supplement(s) for figure 7:

**Figure supplement 1.** Expected value (EV)-representational dissimilarity matrices (RDMs).

**Figure supplement 2.** Univariate fMRI analyses.

can the observed neural and behavioral differences between groups be explained by inter-individual differences in static value synthesis, without caring about self-organized plasticity?

To address this question, we fit the ANN model of value synthesis, with and without self-organized plasticity, to each participant's series of gamble decisions (given the corresponding prospective gains and losses). In what follows, we refer to the models' predictions about fitted behavioral data as models' *postdiction*. We then perform counterfactual model simulations: for each subject-specific fitted model, we simulate the trial-by-trial gamble decisions that would have been observed, had this subject/model been exposed to the sequence of prospective gains and losses that each subject *of the other group* was exposed to (see Methods). That is, we ask what an ANN trained on the gambling decisions of a participant in the narrow gain range group would predict when exposed to the trial-by-trial series of gains and losses of participants from the wide gain range group (and reciprocally). These out-of-sample predictions provide a strong test of the model's generalization ability. *Figure 8* below shows both postdiction and out-of-sample predictions of the two ANN model variants (static versus efficient value synthesis).

Unsurprisingly, both model *postdictions* accurately describe the qualitative group difference in gambling behavior. In addition, both candidate ANNs perform similarly to the logistic model, in terms of both percentage of explained variance (wide range group: 65.8±2.5%, narrow range group: 75.1±1.9%) and balanced fit accuracy (wide range group: 87.3±0.9%, narrow range group: 92.1±0.7%). We note that both ANN models yield postdiction error rates that are similar to the logistic model, in that they are maximal for hard decisions, i.e., when EV lies around zero (see *Figure 8—figure*

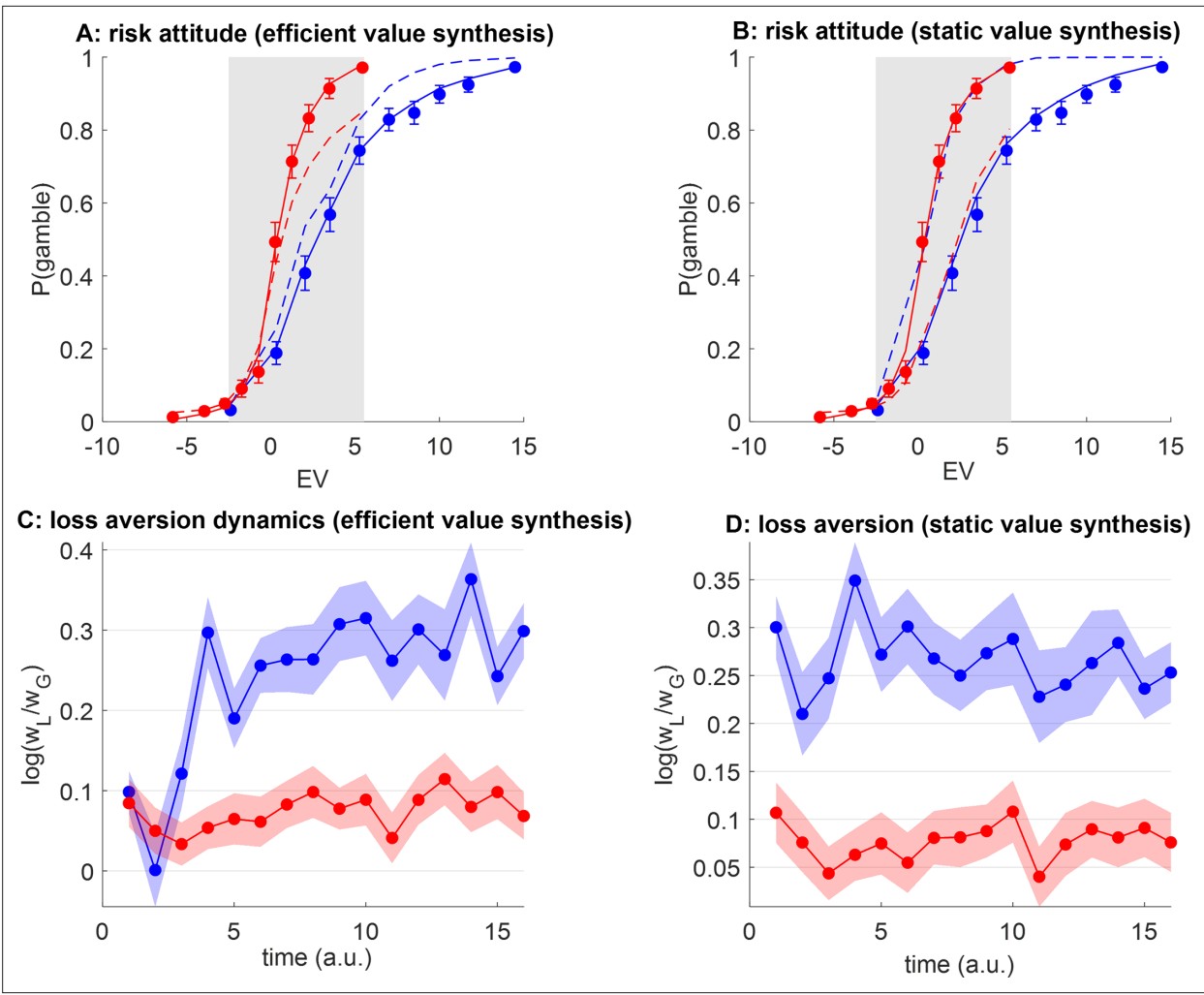

**Figure 8.** Efficient value synthesis: Artificial neural network (ANN) analysis of behavioral data. (**A**) The probability of gamble acceptance under plastic ANNs (y-axis) is plotted against gambles' expected value (EV) (x-axis), in both groups (red: narrow range, blue: wide range). Dots show raw data (error bars depict s.em.). Plain and dashed lines show postdiction and out-of-sample predictions (see main text), respectively. The gray-shaded area highlights the range of expected values that is common to both groups. (**B**) Same as panel A, for static ANN. (**C**) Postdicted loss aversion (y-axis) is plotted as a function of time (x-axis) for both groups (same color code), under the efficient value synthesis model. (**D**) Same as panel C, for static value synthesis.

The online version of this article includes the following figure supplement(s) for figure 8:

**Figure supplement 1.** Artificial neural network (ANNs') postdiction error rate.

**Figure supplement 2.** Self-organized plasticity parameter estimates.

*supplement 1*). In addition, under the efficient value synthesis scenario, empirical distributions of self-organized plasticity parameter estimates are comparable across both groups of participants (see *Figure 8—figure supplement 2*). But did ANN models capture a mechanism that faithfully generalizes to different gain range contexts (i.e. across groups)? First, static ANNs do not yield accurate out-of-sample predictions. This is expected, because static ANNs cannot exhibit range adaptation. Thus, they leave gambling behavior unchanged within the common range of expected values, and simply extrapolate postdicted behavior outside that range (as is the case for the logistic model, see *Figure 4—figure supplement 1*). In other terms, within the range of expected values that is common to both groups, static ANNs wrongly predict that the gambling rate should be higher in the wide range group than in the narrow range group (mean gambling rate group difference=9.7%). The situation is quite different for plastic ANNs, which yield more accurate out-of-sample predictions of peoples' risk attitudes within the common range of expected values. In particular, plastic ANNs correctly predict that gambling rate should be lower in the wide-range group than in the narrow-range group (mean

gambling rate group difference=−6.2%, p=0.038, *F*=1.8). We then measured the absolute out-of-sample prediction error of both plastic and static ANN models, for both participant groups. We found that it was significantly greater for static than for plastic ANN models (wide range group: p=0.013, *F*=9.45, narrow range group: p=0.032, *F*=6.43).

We also quantified postdicted loss aversion dynamics under both types of models (*Figure 8C and D*). One can see that ANNs that operate efficient value synthesis do exhibit realistic loss aversion dynamics, whereby both groups are initially comparable and then progressively spread apart as time unfolds (and the impact of the range of prospective gains accumulates). Note that this systematic dynamical change in peoples' behavior is the information that plastic ANNs exploit to calibrate both the magnitude and the rate of self-organized plasticity, which reacts to the past history of prospective gains and losses. This does not hold, however, for ANNs that operate static value synthesis, which overlook dynamical changes and attempt to explain gambling choices in terms of an idiosyncratic value landscape. Note that, under the efficient value synthesis scenario, the dynamics of self-organized plasticity are determined by magnitude ($\alpha$ in *Equation 2*) and rate ($\beta$ in *Equation 2*) parameters. Accordingly, inter-individual differences in fitted plasticity magnitudes -but nor plasticity rates- significantly correlate with inter-individual differences in behavioral loss aversion indices (narrow range: p=0.024, wide range: p<10$^{-3}$, see *Figure 8—figure supplement 2*). Taken together, these results suggest that the self-organized plasticity mechanism in *Equation 2* is necessary to capture the context-dependency of peoples' loss aversion.

We now aim to evaluating the neurophysiological validity of fitted ANN models of value synthesis. To address this question, we ask whether the activity patterns in ANN models that were fitted to each participant's gambling choices resemble the corresponding within-subject fMRI activity patterns in the

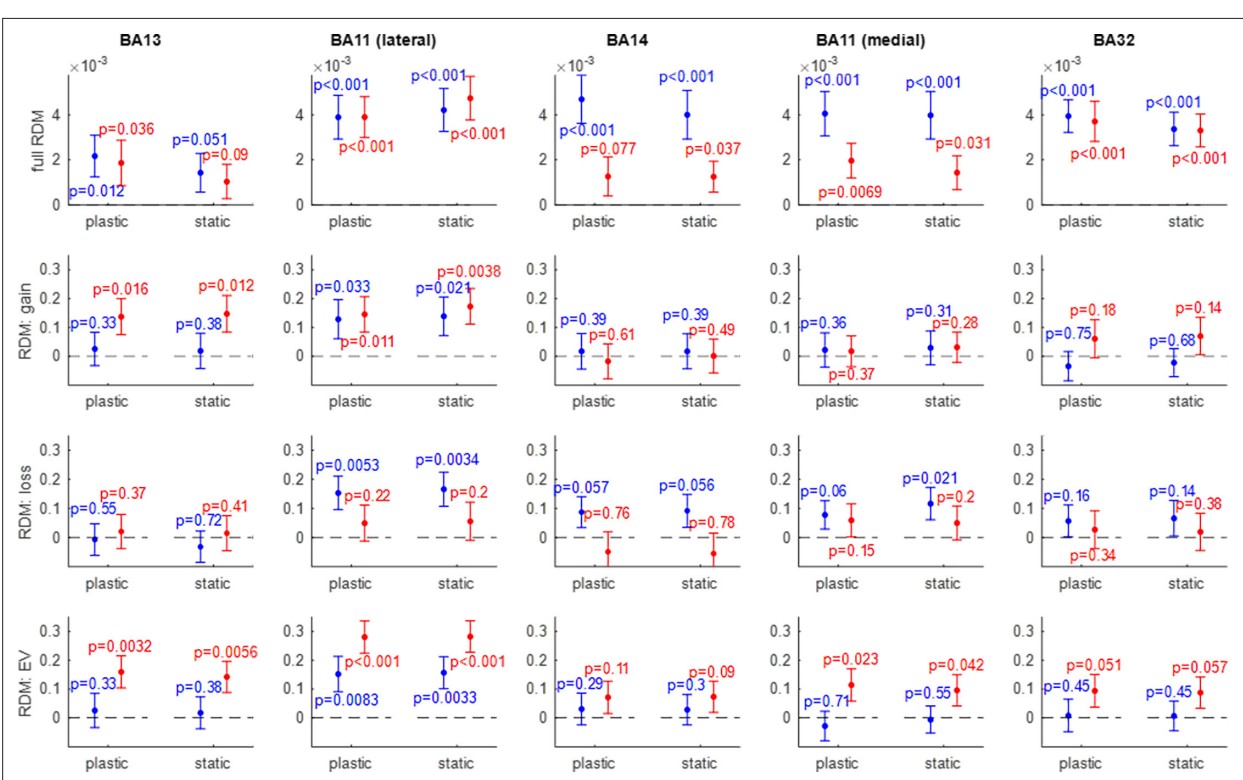

**Figure 9.** Efficient value synthesis: Representational similarity analysis (RSA) analysis results. Within each panel, the correlation between artificial neural network (ANN)-based and fMRI-based representational dissimilarity matrices (RDMs) (y-axis) is shown for both groups of participants (red: narrow range group, blue: wide range group), and both ANN variants (left: plastic ANN, right: static ANN). Errorbars depict within-group s.e.m., and p-values are uncorrected for multiple comparisons. Each column shows the representational similarity analysis (RSA) results of a given OFC subregion from left to right: BA13 (medial), BA11 (lateral), BA14 (rostral), BA11 (medial), and BA32 (rostral). Each row shows one type of RDM (from top to bottom: trial-by-trial RDMs, gain-RDMs, loss-RDMs, and expected value (EV)-RDMs).

The online version of this article includes the following figure supplement(s) for figure 9:

**Figure supplement 1.** Response profile diversity in HP-artificial neural networks (ANNs') integration units.

OFC. We approach this problem using representational similarity analysis (RSA) within each subregion of the OFC. This allows us to compare the trial-by-trial multivariate activity patterns of candidate ANNs with those of fMRI signals in the OFC, without any additional ANN parameter adjustment. In brief, we compute four types of within-subject Representational Dissimilarity Matrices or RDMs (see Methods): (i) full trial-by-trial RDMs, (ii) gain-RDMs, where trials have been binned according to prospective gains, (iii) loss-RDMs, and (iv) EV-RDMs. We then measure the correlation between ANN-based and fMRI-based RDMs, for each OFC subregion and each participant. We then test for the statistical significance of this correlation within each group of participants (one-sample t-tests on Fisher-transformed within-subject correlations). *Figure 9* below summarizes the RSA results in terms of the group-average RDM correlations, for both plastic and static ANNs.

Intriguingly, both plastic and static ANN variants yield trial-by-trial activity patterns that significantly correlate with fMRI activity patterns in almost all OFC subregions (upper panels in *Figure 9*). This suggests that raw fMRI estimates of trial-by-trial dynamics of neural activity are not reliable enough to reveal the functional segregation of OFC subregions. This is not the case, however, when considering gain/loss/EV-RDMs. In brief, irrespective of the ANN model variant, no OFC subregion reaches statistical significance in both groups, for all types of RDM. Nevertheless, the lateral part of Brodmann area 11 almost meets this criterion, in that all RSA analyses are significant except for loss-RDMs in the narrow range group of participants, for both ANN variants (plastic ANN: p=0.22, static ANN: p=0.20). Given the anatomical specificity of this result, this is strong evidence that ANNs that operate value synthesis (whether plastic or static) provide a reasonably realistic prediction of the representational geometry of OFC neurons within the lateral part of Brodmann area 11. We note that, irrespective of the type of RDM considered, nowhere in the OFC is the comparison between the two model variants statistically significant.

Interestingly, ANNs that operate efficient value synthesis also reproduce other known features of value-coding neurons in the OFC. Recall that OFC neurons are notoriously diverse in their response profile, but a consistent finding is that, in the context of value-based decision-making, they can be classified in terms of so-called 'choice cells,' 'chosen value cells,' and 'offer value cells' (*Padoa-Schioppa and Assad, 2006*; *Padoa-Schioppa and Assad, 2008*). Given that this can be considered a pre-requisite for any computational model of value coding in the OFC, we asked whether plastic ANNs reproduce this known property of OFC neurons. For each subject, we thus tested whether the response of integration units correlates (across trials) with choice, chosen value, and/or gamble value, where value is defined as the weighted sum of gains and losses (according to the static logistic model parameter estimates). The results of this analysis are shown on *Figure 9—figure supplement 1*: in brief, plastic ANNs do exhibit this type of apparent coding variability, and predicted category proportions are qualitatively comparable to those reported in the existing literature. This provides additional neurobiological validity to ANN models of efficient value synthesis in the OFC.

Finally, we show that other adaptation models (in particular: efficient coding at the level of decision attributes) cannot explain neural data on value range adaptation. This is summarized in *Figure 10* below. Although they predict qualitatively similar behavioral range adaptation effects (see *Figure 10—figure supplement 1*), they do not predict value range adaptation in the ANN's integration layer (*Figure 10AB*). They also predict that increasing the range of prospective gains should *increase* the neural encoding strengths of gains within the integration layer (*Figure 10C*), which is at odds with the empirical data that we report here (*Figures 5 and 6*). Finally, when fitted on participants' behavioral choices, they do not generalize well across gain range contexts (*Figure 10—figure supplement 2*), and their RSA results are less convincing (even in the lateral part of Brodman area 11, see *Figure 10—figure supplement 3*). The mathematical derivation of such models, as well as the analysis of their predictions, are summarized in the Supplementary material.

Taken together these behavioral and neural analyses provide converging evidence that self-organized plasticity that operates efficient value synthesis in the lateral part of BA11 is a likely explanation for range adaptation of loss aversion.

## Discussion

In this work, we investigate the neural range adaptation mechanism in OFC neurons that underlies the irrational context-dependency of value-based decisions. We focus on risky decisions, where value needs to be constructed out of primitive decision attributes (here: prospective gains and losses). This

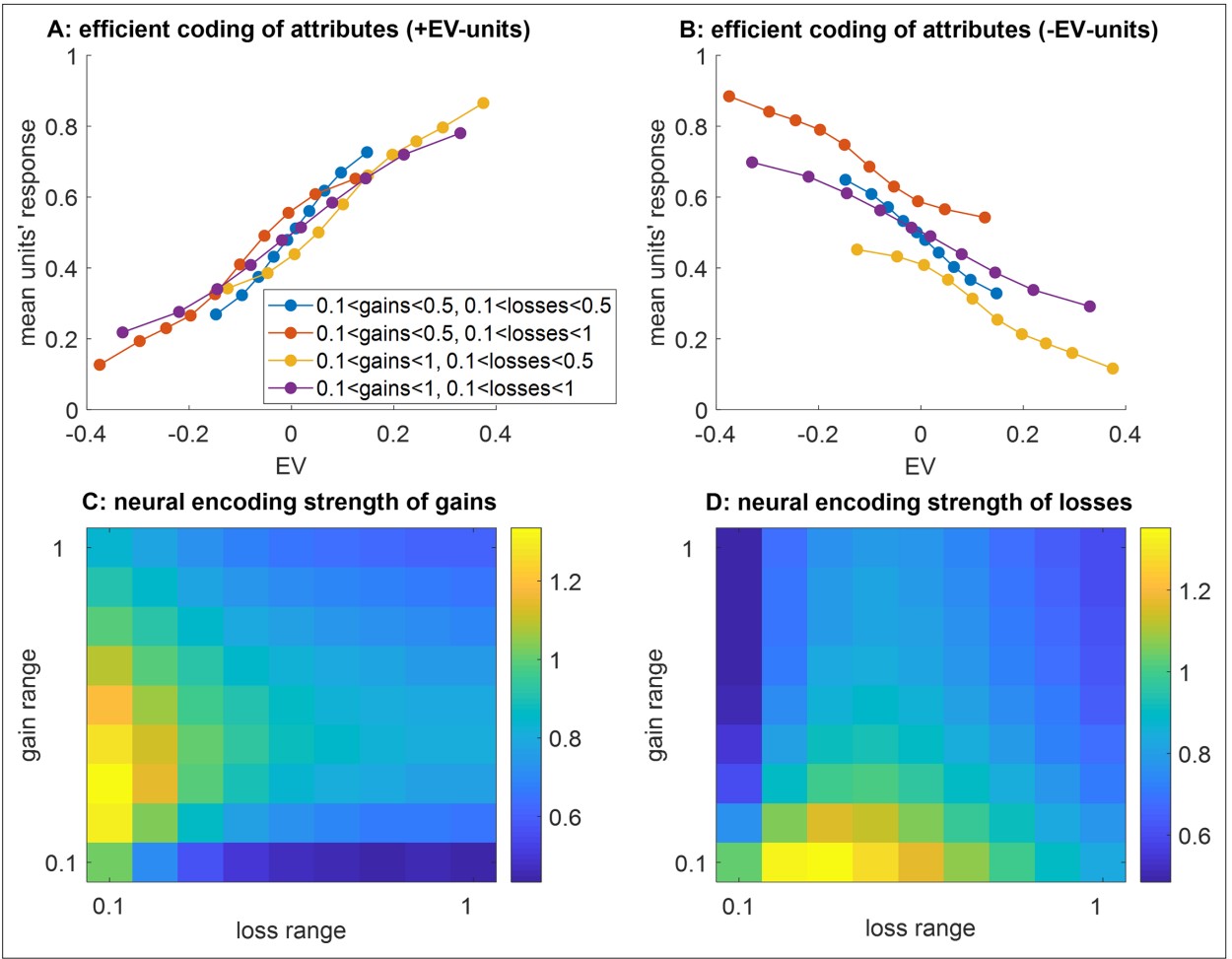

**Figure 10.** Main divergent predictions of the *efficient coding of attributes* scenario. (**A**) The average units' response (y-axis) to pairs of prospective gains and losses that fall within predefined expected value (EV) bins (x-axis) is shown, while ANNs that operate efficient value synthesis are exposed to four different spanned gain/loss domains (blue: narrow ranges of gains and losses, violet: wide ranges of gains and losses, red: narrow gain range and wide loss range, yellow: wide gain range and narrow loss range, see main text). Only integration units that correlate positively with EV are shown. (**B**) Integration units that correlate negatively with EV, same format as panel (**A**). (**C**) The neural encoding strength of gains (color code: from blue -low encoding strength- to yellow -high encoding strength-) is shown as a function of the spanned range of losses (x-axis, range increases from left to right) and gains (y-axis, range increases from bottom to top). (**D**) Neural encoding strength of losses, same format.

The online version of this article includes the following figure supplement(s) for figure 10:

**Figure supplement 1.** Efficient coding of attributes: impact of spanned ranges of gains and losses.

**Figure supplement 2.** Efficient coding of attributes: Artificial neural network (ANN) analysis of behavioral data.

**Figure supplement 3.** Efficient coding of attributes: Representational similarity analysis (RSA) analysis results.

eventually disambiguates the neural and behavioral implications of candidate computational scenarios for range adaptation. We show that a specific form of self-organized plasticity between attribute-specific and attribute-integration neurons best predicts (out-of-sample) both context-dependent behavioral biases and range adaptation in OFC neurons.

The processing of reward signals in OFC neurons is known to exhibit range adaptation (*Conen and Padoa-Schioppa, 2019*; *Louie and Glimcher, 2012*; *Padoa-Schioppa, 2009*; *Rangel and Clithero, 2012*). The typical explanation is that OFC neurons adapt their output firing properties to match the recent history of values (*Polanía et al., 2019*). Implicit in this reasoning is the idea that OFC neurons are receiving value signals, which they are transmitting to downstream decision systems (*Padoa-Schioppa and Rustichini, 2014*; *Rustichini et al., 2017*). However, this assumption is at odds with the notion that OFC neurons are rather constructing value from input signals about decision-relevant attributes (*O'Doherty et al., 2021*; *Pessiglione and Daunizeau, 2021*; *Raghuraman and*

*Padoa-Schioppa, 2014*). An important contribution of this work is to show that such a value synthesis scenario is compatible with known value range adaptation effects in OFC neurons (*Figure 2*). In particular, our results suggest that value range adaptation may be the byproduct of self-organized plasticity that aims at mitigating information loss induced by limited neural response ranges. At the behavioral level, this scenario predicts that peoples' sensitivity to decision attributes inversely scales with the range of each decision attribute. In the context of gamble decisions, this implies that loss aversion follows the ratio of spanned ranges of gain w.r.t. losses (*Figure 3*). This systematic dependence of peoples' loss aversion on the relative ranges of spanned gains and losses has already been documented (*Rakow et al., 2020*). However, when considering behavioral data alone, the interpretative power of this kind of experimental design is limited (*Williams et al., 2021*).

In fact, the same behavioral pattern can be predicted under simpler efficient coding scenarios that operate at the level of attribute-specific layers (see the section on 'efficient coding of gains and losses' in the Supplementary materials). Interestingly, these models make neural predictions that are distinct from the efficient value synthesis scenario. In particular, ANNs that operate efficient coding of attributes do not exhibit value range adaptation effects in integration units (see *Figure 10A, B*). Also, they predict that the neural encoding strength of gains in integration units increases when the spanned range of gains increases (see *Figure 10C*). This clearly goes against the results of our model-free analyses of fMRI data in the OFC. The distinction between these two scenarios (i.e. efficient coding of attributes versus efficient value synthesis) is important, because it may confound the relationship between range adaptation in OFC neurons and its behavioral consequences. In this sense, our results complement and extend previous computational modeling studies that focus on the behavioural impact of range adaptation in attribute-specific units (*Soltani et al., 2012*). In principle, the two mechanisms may coexist. Importantly, however, range adaptation within the attribute layers does not obviate the need for range adaptation within the integration layer. This is simply because each integration unit receives an arbitrary mixture of inputs. More generally, within a hierarchical system relying on units equipped with saturating activation functions, efficient information processing requires range adaptation at all levels of the hierarchy. Having said this, many other candidate neural mechanisms may, in principle, compete or interact with the self-organized plasticity that we have considered here, eventually crystalizing or destabilizing plastic changes. This is the case for, e.g., Hebbian and homeostatic plasticities, which are known to induce slow neural hysteretic effects (*Fox and Stryker, 2017*; *Pezzulo et al., 2015*; *Toyoizumi et al., 2014*; *Turrigiano, 2017*). Recent theoretical arguments also suggest that flexible attribute integration in OFC neurons may necessitate plastic changes in the synaptic gain of upstream attribute-specific neurons (*O'Doherty et al., 2021*). More precisely, the wiring between attribute-specific and attribute-integration neurons should self-organize according to the contextual relevance of attributes. The extent to which the properties of these or similar kinds of neurophysiological mechanisms may explain contextual dependence and/or irrational behavioral responses is an open and challenging issue.

At the neural level, it is reassuring to see that fMRI patterns of activity in the lateral part of Brodmann area 11 strongly resemble the quantitative predictions of plastic ANN models. One might find it disappointing that these predictions turn out not to be verified in Brodmann area 32, owing to the known value encoding within the ventromedial PFC (see, e.g. *Clairis and Pessiglione, 2022*; *Lopez-Persem et al., 2020*). In fact, there is an ongoing debate regarding the relative contribution of OFC subregions w.r.t. value processing. For example, lateral, but not medial, OFC may host representations of attributes that presumably compose value judgements (*Suzuki et al., 2017*). Although this clearly aligns with our model-free fMRI data analysis results, we do not claim that the evidence we provide regarding the anatomical location of value synthesis generalizes beyond decision contexts that probe peoples' loss aversion. In fact, our main claim is about whether and how efficient value synthesis operates within the OFC, as opposed to which specific subregion of the OFC drives the adaptation of loss aversion and/or related behavioral processes.

Having said this, we note that ANN integration units do exhibit response profiles that are reminiscent of typical OFC neurons electrophysiological activity during value-based decision-making. For example, they reproduce the diversity of coding that has been repeatedly observed in OFC neurons during value-based decision-making ('offer value cells,' 'chosen value cells,' and 'choice cells;' *Padoa-Schioppa and Assad, 2006*, *Padoa-Schioppa and Assad, 2008*). This is summarized in *Figure 9—figure supplement 1*. We see this as a non-specific byproduct of the mixed selectivity of integration

units, which exhibit arbitrary complex and idiosyncratic receptive fields (see *Figure 1—figure supplements 1 and 3*). More importantly, integration units also exhibit the known properties of value range adaptation in these same neurons (*Figure 2*). Intriguingly, however, value range adaptation in 'offer value cells' had been observed without any significant behavioral preference change. Under the assumption that preferences between offers derive from the direct comparison of output signals from distinct 'offer value cells,' this is surprising. To solve this puzzle, later theoretical work proposed that value range adaptation is somehow 'undone' downstream value coding in the OFC (*Padoa-Schioppa and Rustichini, 2014*). In our context, this would suggest that readout weights ($w^{(k)}$ in *Equation 2*) would compensate for value-related adaptation, effectively thwarting the behavioral consequences of self-organized plasticity between attribute and integration layers. However, this reasoning critically relies upon the assumed computational role of 'offer value cells.' In fact, this puzzle may effectively dissolve under other scenarios of how 'offer value cells' contribute to decision-making. Recall that this null result was obtained in a decision context where choice options were characterized in terms of the type of offer (i.e. juices that differ w.r.t. palatability), whose quantity was systematically varied. Here, value synthesis would effectively aggregate two attributes, namely palatability and quantity. Under this view, 'offer value cells' simply are integration units that show a certain form of mixed selectivity, whereby units' sensitivity to quantity strongly depends upon palatability. At this point, one needs to consider candidate scenarios of how the OFC may operate value synthesis for multiple options in a choice set. A possibility is that the OFC is automatically computing the value of the option that is currently under the attentional focus (*Lebreton et al., 2009*; *Lopez-Persem et al., 2020*), while storing the value of previously attended options within an orthogonal population code (*Pessiglione and Daunizeau, 2021*). In principle, this implies that the OFC is wired such that it can handle arbitrary switches in attentional locus without compromising the integration of option-specific attributes. In this scenario, integration units (including those that look like 'offer value cells') would adapt to the range of all incoming attribute signals, irrespective of which option in the choice set is currently attended. In turn, 'offer value cells' would look like they are only partially adapting to the value range of a given offer type (*Burke et al., 2016*; *Conen and Padoa-Schioppa, 2019*). More importantly, to the extent that between-attribute spillover effects are negligible, changes in the range of offer quantities would distort the readout value profile along the quantity dimension without altering the palatability dimension. This would effectively leave the relative preference between offer types unchanged. Of course, this is only one candidate scenario among many. Nevertheless, we would still argue that the behavioral consequences of range adaptation in 'offer value cells' actually depend upon their underlying computational role.

Now, whether this sort of ANN model produces 'realistic' electrophysiological activity profiles beyond this kind of empirical observation is questionable. The reason is twofold. First, they are agnostic w.r.t. within-trial temporal dynamics. Second, there is some level of arbitrariness in the modeling assumptions (e.g. ANN structural constraints) that cannot be finessed using either behavioral or neuroimaging data. What we argue is robust in these ANN models is the information content that they carry, which is distributed over the activity profiles of their artificial neural unit layers. This is the main reason why we resort to variants of RSA analyses for comparing their predictions to multivariate fMRI activity patterns.

At this point, let us comment on a seemingly innocuous neural modeling assumption: namely, that units' input-output activation functions are saturating. This was motivated by the fact that neurons' firing rate cannot exceed some predefined physiological limit (see, e.g. *Wang et al., 2016*). Under the framework of efficient coding, such response range limitation is eventually what creates the need for range adaptation. This is because information loss mostly follows from inputs reaching the saturating domain of units' activation functions. However, one may wonder how robust our efficient value synthesis scenario to deviations from this assumption is. Analytical derivations show that other monotonic (and bounded) activation functions would yield very similar self-organized plasticity rules. This means that our results would generalize to any monotonic activation function. However, it turns out that efficient value synthesis yields unstable self-organized plasticity dynamics under non-monotonic (e.g. Gaussian or bell-shaped) activation functions. To understand this, recall that the self-organized plasticity rule derives from aligning the connectivity change with the gradient of information loss w.r.t. connection strengths. This gradient explodes when inputs fall within domains where the derivative of the activation function approaches zero. This unavoidably happens with non-monotonic activation

functions because the plasticity mechanism eventually focuses the weighted inputs within the vicinity of their mode. In other terms, one may argue that only monotonic activation functions are compatible with the efficient value synthesis scenario.

Now, how generalizable is the neural mechanism we disclose here? We argue that self-organized plasticity may explain many forms of persistent irrational behavioral changes, through gradual range adaptation effects in OFC neurons. We note that, in our context, these changes seem to unfold over several minutes (*Figure 4D*), which is consistent with the fastest time scale of long-term potentiation/depression (*Abraham, 2003*). However, we contend that the evidence we provide here is insufficient to establish whether these changes remain stable over longer periods and whether they can be overcome by explicit instructions or intensive training (*Cicchini et al., 2012*). A related issue is whether similar plasticity mechanisms may explain virtually instantaneous range adaptation in value-coding neurons (*Louie et al., 2015*), eventually driving behavioral phenomena such as the framing effect. Here, we speculate that the framing of decisions may automatically trigger contextual expectations regarding expected gain and/or loss ranges, which may induce fast plastic changes within value-constructing networks through, e.g., short-term potentiation (*Fiebig and Lansner, 2017*).

That the brain's biology is to blame for all kinds of cognitive and/or behavioral flaws is not a novel idea (*Buschman et al., 2011*; *Marois and Ivanoff, 2005*; *Miller and Buschman, 2015*; *Ramsey et al., 2004*). However, providing neuroscientific evidence that a hard-wired biological constraint shapes and/or distorts the way the brain processes information is not an easy task. This is because whether the brain deviates from how it *should* process a piece of information is virtually unknown. This is particularly true for value-guided decision-making, which relates to subjective assessments of preferences rather than objective processing of decision-relevant evidence (*Rangel et al., 2008*). Nevertheless, value-guided decision-making is known to exhibit many irrational biases, the neurocognitive explanations of which have been the focus of intense research over the past decades. From a methodological standpoint, our main contribution is to show how to leverage computational models (in particular: ANNs) to test hypotheses regarding neurophysiological mechanisms that may constrain or distort behaviorally-relevant information processing. On the one hand, we retain the simplicity of established 'model-based' fMRI approaches (*Borst et al., 2011*; *O'Doherty et al., 2007*), which proceed by cross-validating the identification of hidden computational determinants of behavior with neural data. On the other hand, our dual ANN/RSA approach enables us to quantify the statistical evidence for neurophysiological mechanisms that are difficult –if not impossible- to include in computational models that are defined at Marr's *algorithmic* level (*McClamrock, 1991*), e.g., normative models of behavior (as derived from, e.g. learning or decision theories) and/or cognitive extensions thereof. Self-organized plasticity between attribute-specific and attribute-integration units is a paradigmatic example of what we mean here. More generally, hard-wired biological mechanisms or constraints may not always be instrumental to the cognitive process of interest. In turn, it may be challenging to account for incidental biological disturbances of neural information processing, when described at the algorithmic level. A possibility here is to conceive of these disturbances as some form of random noise that perturbs cognitive computations (*Drugowitsch et al., 2016*; *Wyart and Koechlin, 2016*). In contrast, we rather suggest relying on computational models that solve a well-defined computational problem (here: constructing the gambles' subjective value from prospective gains and losses) but operate at the neural level. Accounting for possibly incidental, biological constraints and/or hard-wired mechanisms then enables comparing quantitative/deterministic scenarios for sub-optimal disturbances of covert cognitive processes of interest.

## Methods

### Artificial neural network models of value synthesis

Artificial neural networks or ANNs decompose a possibly complex form of information processing in terms of a combination of very simple computations performed by connected 'units,' which are mathematical abstractions of neurons. Here, we take inspiration from a growing number of studies that use ANNs as mechanistic models of neural information processing (*Güçlü and van Gerven, 2015*; *Kietzmann et al., 2017*; *Kietzmann et al., 2019*; *Kriegeskorte and Golan, 2019*), with the added requirement that they eventually explain (possibly irrational) behavioural data.

In abstract terms, any decision can be thought of as a cognitive process that transforms some input information $u = \left\{ u^{(1)}, u^{(2)}, ..., u^{(n_u)} \right\}$ into a behavioral output response $r$. Here, participants have to accept or reject a risky gamble composed of a 50% chance of obtaining a gain G and a 50% chance of experiencing a loss L, i.e., $u$ is composed of $n_u = 2$ input attributes: $u = \{G, L\}$. Under an ANN model of such decisions, people's behavioral response is the output of a neural network that processes the attributes $u$, i.e.: $r \approx g_{ANN}(u, \vartheta)$, where $\vartheta$ are unknown ANN parameters and $g_{ANN}()$ is the ANN's input-output transformation function. So-called 'shallow' ANNs effectively reduce $g_{ANN}()$ to a combination of neural units organized in a single hidden layer. In what follows, we rather rely on (moderately) deep ANNs with two hidden layers: namely, an attribute-specific layer (which is itself decomposed into gain-specific and loss-specific layers) and an integration layer (which receives inputs from both attribute-specific layers). The units that compose the latter then collectively determine gamble decisions by integrating prospective gains and losses.

We assume that each attribute $u_t^{(i)}$ is encoded into the activity of neurons $\left[ x_t^{(i,1)}, x_t^{(i,2)}, ..., x_t^{(i,j)}, ..., x_t^{(i,n_x)} \right]$ of its dedicated 'attribute-specific layer,' where $n_x$ is the number of attribute-specific neurons per attribute. What we mean here is that the neuron $j$ in the attribute-specific layer $i$ responds to $u_t^{(i)}$ as follows:

$$x_t^{(i,j)} = f_x^{(i,j)} \left( u_t^{(i)}, \theta^{(i,j)} \right) \tag{3}$$

where $f(.)$ is the activation function of neural units that compose the ANN's attribute-specific layer:

$$f(u, \theta) = \frac{1}{1 + exp\left( \frac{\mu - u}{\sigma} \right)} \tag{4}$$

which yields a sigmoidal transform of inputs. Critically, such activation functions are bounded, i.e., we assume that neural units cannot fire beyond a certain rate. In neocortical neurons, the main mechanistic constraint that acts on firing rate is the action potential's refractory period, which depends upon how long it takes ion channels to complete a whole voltage activate-deactivate cycle (about a few milliseconds). Typically, even fast-spiking neocortical neurons cannot fire at a frequency higher than about 500 or 600 Hz (**Wang et al., 2016**). As we will see, this type of response saturation is a critical component of range adaptation.

The parameters $\theta^{(i,j)} = \left\{ \mu^{(i,j)}, \sigma^{(i,j)} \right\}$ in **Equation 3** captures idiosyncratic properties of the neuron $j$ in the input layer $i$ (e.g. its firing rate threshold $\mu^{(i,j)}$ and the slope parameter $\sigma^{(i,j)}$).

Collectively, the activity vector $\left[ x_t^{(i,j)} \right]_{j=1,...,n_x}$ forms a multivariate representation of attribute $u_t^{(i)}$ in the form of a population code (**Ebitz and Hayden, 2021**). Then the output of the attribute-integration layers is passed to the 'integration layer' $\left[ z_t^{(1)}, z_t^{(2)}, ..., z_t^{(k)}, ..., z_t^{(n_z)} \right]$, i.e., the neuron $k$ of the integration layer responds to $\left[ x_t^{(i,j)} \right]_{j=1,...,n_x}^{i=1,...,n_u}$ as follows:

$$\begin{cases} z_t^{(k)} = f_z^{(k)} \left( v_t^{(k)}, \phi^{(k)} \right) \\ v_t^{(k)} = \sum_{i=1}^{n_u} \sum_{j=1}^{n_x} C^{(i,j,k)} x_t^{(i,j)} \end{cases} \tag{5}$$

where $C^{(i,j,k)}$ is the connection weight from the neuron $j$ in the attribute-specific layer $i$ to the neuron $k$ of the integration layer, $\phi^{(k)}$ capture idiosyncratic properties of the integration neuron $k$ (i.e. its firing rate threshold and slope), and $v_t^{(k)}$ are the inputs of integration units.

Collectively, integration neurons form a representation of decision value in the form of a population code, i.e., the gambles' subjective value $V_t$ at the time or trial $t$ is read out from the integration layer as follows:

$$V_t = \sum_{k=1}^{n_z} W^{(k)} z_t^{(k)} \tag{6}$$

where $W^{(k)}$ are the population readout weights.

Taken together, *Equations 3–6* define the end-to-end ANN's transformation of prospective gains and losses into decision value $V_t = V(u_t, \vartheta)$:

$$V(u_t, \vartheta) \triangleq \sum_{k=1}^{n_z} W^{(k)} f_z \left( \sum_{i=1}^{n_u} \sum_{j=1}^{n_x} C^{(i,j,k)} f_x \left( u_t^{(i)}, \theta^{(i,j)} \right), \phi^{(k)} \right) \tag{7}$$

where $\vartheta$ lumps all ANN parameters together, i.e.,: $\vartheta \triangleq \{W, C, \theta, \phi, \upsilon\}$. This is what we coin *value synthesis*. A schematic summary of the ANN's double-layer structure is shown on panel A of *Figure 1*.

## Efficient value synthesis and self-organized plasticity

We start with the premise that the brain system that integrates value-relevant option features (here: prospective gains and losses) to construct value signals may be doing this under neural noise, which degrades the information about value. In particular, the limited range of physiological responses of neural units that perform this integration induces some information loss on value signals. This is because, when inputs to integration units fall too far away from their firing threshold (say outside a $\pm 2\sqrt{2}\sigma$ range), activation functions saturate, i.e., they produce non-discriminable outputs (close to 0 or 1). In this context, *efficient value synthesis* refers to the idea that neural networks that perform the integration of prospective gains and losses to construct value may adapt their response properties to mitigate information loss, hence the 'efficiency' of value synthesis. We now sketch how *efficient* value synthesis can be achieved within ANNs whose 2-layer structure is described in *Equations 3-6*.

In the presence of neural noise, the ANN's readout value $\widetilde{V}_t$ of a gamble made of a pair $(G_t, L_t)$ of prospective gain and loss is given by (in lieu of *Equation 6*):

$$\begin{cases} \widetilde{V}_t = \sum_k w^{(k)} \widetilde{z}_t^{(k)} \\ \widetilde{z}_t^{(k)} = z^{(k)} (G_t, L_t) + \eta_t^{(k)} \end{cases} \tag{8}$$

where $\eta_t^{(k)}$ is some (uncontrollable) neural noise that competes with the 'utile' component $z^{(k)} (G_t, L_t)$ that is given in *Equation 5*.

*Equation 8* can serve to measure the information loss $IL$ that is induced by neural noise under units' limited response range:

$$IL = -MI \left( \widetilde{z}, z \right) \underset{\eta \to 0}{\to} K - H[z] = K - H[\upsilon] - \sum_k E \left[ ln \left| \frac{\partial f_z^{(k)}}{\partial \upsilon^{(k)}} \right| \right] \tag{9}$$

Equation 9 states that the information loss increases when the mutual information between the noisy responses of integration units and their "utile" (i.e. noiseless) component decreases. Here, $MI(\cdot, \cdot)$ is Shannon's mutual information, $K$ is a constant, $H[\cdot]$ is Shannon's entropy, and $E[\cdot]$ is the standard expectation operator. The right-hand term in *Equation 9* arises at the small noise limit (*Nadal, 1994*), and the expectation is taken under the distribution of integration units' inputs $\upsilon$. The last term in *Equation 9* is simply the average steepness (in log space) of units' activation functions. Importantly, *Equation 9* holds irrespective of the type of nonlinearity of ANN units' activation functions.

The entropy $H[\upsilon]$ has no closed-form expression, but can be given a multivariate gaussian approximation, i.e.,: $H[\upsilon] \approx ln \left| CSC^T \right| /2 + K'$, where $S = E \left[ xx^T \right]$ is the covariance matrix of the output of the ANN's first layer and $K'$ is a constant. In principle, this approximation works because, when the size of the network grows, the central limit theorem implies that the distribution of integration units' inputs $\upsilon$ will tend towards normality. The robustness of this approximation has been established in the context of undercomplete ICA (*Porrill and Stone, 1998*).

Efficient value synthesis can then be simply achieved by modifying the connectivity matrix $C$ to decrease the information loss $IL$, i.e., along the direction of the information loss gradient:

$$\Delta C = -\alpha \frac{\partial IL}{\partial C} = \alpha \frac{\partial H[v]}{\partial C} + \alpha \sum_k \frac{\partial}{\partial C} E\left[ln\left|\frac{\partial f_z^{(k)}}{\partial v^{(k)}}\right|\right] \tag{10}$$

where $\alpha$ controls the magnitude of the gradient-following step and the first term in the right-hand side of *Equation 10* is given by:

$$\frac{\partial H[v]}{\partial C} \approx \underbrace{(CSC^T)^{-1}CS}_{non\ local} \tag{11}$$

The matrix multiplier in the right-hand side of *Equation 11* is non-local, i.e., the gradient $\partial H[v]/\partial C^{(i,j,k)}$ depends upon all connection weights in the network. This is unrealistic for biological systems, and we thus drop this term in the remainder of this manuscript. In turn, *Equation 11* can be approximated as a collection of local changes to the connectivity matrix:

$$\Delta C^{(i,j,k)} \approx \alpha \frac{\partial}{\partial C^{(i,j,k)}} E\left[ln\left|\frac{\partial f_z^{(k)}}{\partial v^{(k)}}\right|\right] \tag{12}$$

We will see that *Equation 12* only involves the output response $z^{(k)}$ and $x^{(i,j)}$ of the pair of attribute and integration units that are connected through $C^{(i,j,k)}$. *Equation 12* implies that efficient integration will tend to change the distribution of inputs $v^{(k)}$ to each integration unit such that they span the range where the steepness of its activation function is maximal. Focusing inputs to the responsive range of integration units' activation functions then maximizes the output variability induced by attribute inputs. This makes sense, since this is expected to yield maximal contrast over the response outputs of integration units.

But *Equation 12* still requires a last modification to derive a realistic self-organized plasticity rule for efficient value synthesis. This is because self-organized plasticity is a dynamical process, which reacts to recent network activity, as trials and/or time unfolds.

Note that the expectation in *Equation 12* is taken under the distribution of prospective gains and losses, and can, therefore, be defined as a sample average over trial-by-trial gamble attributes. If the underlying distribution is non-stationary, then $E[\cdot]$ can be estimated at trial or time $t$ using a simple weighted moving average operator $\hat{E}_t[\cdot]$:

$$\hat{E}_t\left[ln\left|\frac{\partial f_z^{(k)}}{\partial v^{(k)}}\right|\right] = \beta \sum_{t'=1}^{t} (1-\beta)^{t-t'} ln\left|\frac{\partial f_z^{(k)}}{\partial v^{t'(k)}}\right| = (1-\beta)\ \hat{E}_{t-1}\left[ln\left|\frac{\partial f_z^{(k)}}{\partial v^{(k)}}\right|\right] + \beta\ ln\left|\frac{\partial f_z^{(k)}}{\partial v^{t(k)}}\right| \tag{13}$$

where $\beta$ (note: $0 < \beta < 1$) controls the exponential decay of past samples' weights in the moving average operator.

Let $\Delta C_t^{(i,j,k)}$ be the change of connectivity at trial or time $t$. Replacing the expectation in *Equation 12* with the moving average operator $\hat{E}_t[\cdot]$ in *Equation 13* now yields:

$$\Delta C_t^{(i,j,k)} \approx \alpha \frac{\partial}{\partial C^{(i,j,k)}} \hat{E}_t\left[ln\left|\frac{\partial f_z^{(k)}}{\partial v^{(k)}}\right|\right] = \alpha(1-\beta)\Delta C_{t-1}^{(i,j,k)} + \alpha\beta \frac{\partial}{\partial C^{(i,j,k)}} ln\left|\frac{\partial f_z^{(k)}}{\partial v_t^{(k)}}\right| \tag{14}$$

where the local gradient can be written as:

$$\frac{\partial}{\partial C^{(i,j,k)}} ln\left|\frac{\partial f_z^{(k)}}{\partial v^{(k)}}\right| = \left|\frac{\partial f_z^{(k)}}{\partial v^{(k)}}\right|^{-1} \frac{\partial}{\partial v^{(k)}} \left|\frac{\partial f_z^{(k)}}{\partial v^{(k)}}\right| \frac{\partial v^{(k)}}{\partial C^{(i,j,k)}} \tag{15}$$

Under sigmoidal activation functions, then:

$$\begin{cases} \left| \dfrac{\partial f_z^{(k)}}{\partial v^{(k)}} \right| = \dfrac{\partial f_z^{(k)}}{\partial v^{(k)}} = \dfrac{f_z^{(k)}\left(1 - f_z^{(k)}\right)}{\sigma_z(k)} \\[3ex] \dfrac{\partial}{\partial v^{(k)}}\left| \dfrac{\partial f_z^{(k)}}{\partial v^{(k)}} \right| = \dfrac{\partial^2 f_z^{(k)}}{\partial v^{(k)2}} = \dfrac{1}{\sigma_z(k)} \dfrac{\partial f_z(k)}{\partial v(k)}\left(1 - 2 f_z^{(k)}\right) \end{cases} \tag{16}$$

Replacing *Equation 16* into *Equations 14-15* then yields:

$$\Delta C_t^{(i,j,k)} = \alpha \left(1 - \beta\right) \Delta C_{t-1}^{(i,j,k)} + \alpha \frac{\beta}{\sigma_z^{(k)}} \left(1 - 2 z_t^{(k)}\right) x_t^{(i,j)} \tag{17}$$

which only depends upon the output response of connected pairs of attribute-specific and attribute-integration units.

Note that accounting for the nonlocal component of *Equations 10-11* would require inserting the correction term $\alpha\beta \left[ (v_t\, v_t^T)^{-1}\, v_t\, x_t^T \right]_{ij}$ in the right-hand side of *Equation 17*. In our experience, its magnitude is typically small when compared to the Hebbian term in *Equation 17*. In turn, this term can be neglected without altering the main properties of efficient value synthesis.

*Equation 17* states that efficient value synthesis can be operated by local, history-dependent, *self-organized plasticity* within the network. The plasticity in *Equation 17* is 'self-organized' in the sense that it does not require any teaching or feedback signal. In this context, $\beta$ determines the adaptation rate of the network's connectivity to changes in the distribution of prospective gains and losses. Importantly, the anti-Hebbian component of self-organized plasticity generalizes to any nonlinear activation function that is continuous and monotonically increasing. This is not the case, however, for non-monotonic activation functions (e.g. pseudo-gaussian activation functions).

In summary, as long as ANN units have monotonically increasing activation functions, efficient value synthesis can be implemented through some self-organized plasticity rule of the form given in *Equation 17*. It turns out that the self-organized plasticity rule in *Equation 17* essentially modifies the integration units' receptive fields, i.e., their pattern of response to a given pair of prospective gain and loss. This has two main consequences: it changes the information content of the network, and it distorts the readout value. We unpack these two phenomena using numerical simulations, which we report in the two first section of the Supplementary materials. At this point, we simply note that the effect of self-organized plasticity on both the readout value profile and the information content within the integration layer depends upon the ranges of prospective gains and losses that the ANN is exposed to. The neural and behavioral impacts of the shape of the spanned domain of gains and losses are summarized in *Figures 2 and 3* of the Results section.

## Behavioral and fMRI data: Experimental paradigm

In this work, we perform a re-analysis of the NARPS dataset (*Botvinik-Nezer et al., 2019*; *Botvinik-Nezer et al., 2020a*), openly available on https://openneuro.org/; *Poldrack et al., 2013*. This dataset includes two studies, each of which is composed of a group of 54 participants who make a series of decisions made of 256 risky gambles. On each trial, a gamble was presented, entailing a 50/50 chance of gaining an amount G of money or losing an amount L. As in *Tom et al., 2007*, participants were asked to evaluate whether or not they would like to play each of the gambles presented to them (strongly accept, weakly accept, weakly reject, or strongly reject). They were told that, at the end of the experiment, four trials would be selected at random: for those trials in which they had accepted the corresponding gamble, the outcome would be decided with a coin toss and for the other ones -if any- the gamble would not be played. In the first study (hereafter: 'narrow range' group), participants decided on gambles made of gain and loss levels that were sampled from within the same range (G and L varied between 5 and 20 $). In the second study (hereafter: the 'wide range' group), gain levels scaled to double the loss levels (L varied between 5 and 20$, and G varied between 10 and 40$). In both studies, all 16×16=256 possible combinations of gains and losses were presented across trials, which were separated by 7 s on average with some random jitter (min 6, max 10).

MRI scanning was performed on a 3T Siemens Prisma scanner. High-resolution T1-weighted structural images were acquired using a magnetization-prepared rapid gradient echo (MPRAGE) pulse sequence with the following parameters: TR=2530ms, TE=2.99ms, FA=7, FOV=224 × 224 mm,

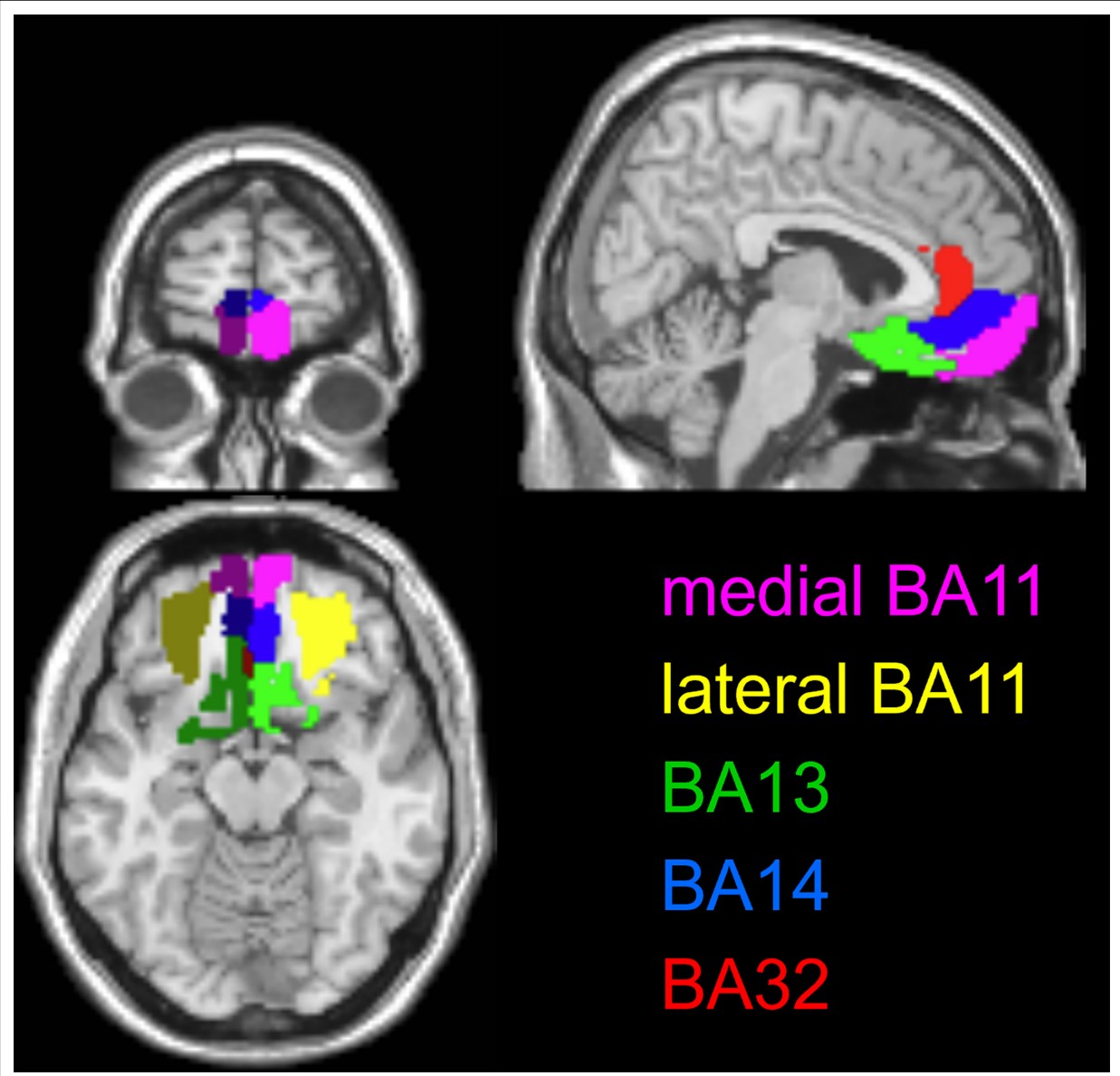

**Figure 11.** Anatomical masks of OFC subregions. Pink: medial part of BA11, yellow: lateral part of BA11, green: A13, blue: BA14, red: BA32. Dark and light colors correspond to left and right hemispheric analogous regions, respectively.

resolution=1 × 1 × 1 mm. Whole-brain fMRI data were acquired using echo-planar imaging with a multi-band acceleration factor of 4 and parallel imaging factor (iPAT) of 2, TR=1000 ms, TE=30 ms, flip angle=68 degrees, in plane resolution of 2 × 2 mm 30 degrees of the anterior commissure-posterior commissure line to reduce the frontal signal dropout, with a slice thickness of 2 mm and a gap of 0.4 mm between slices to cover the entire brain. See https://www.narps.info/analysis.html#protocol for more details.

## Extraction of trial-by-trial BOLD responses within OFC subregions

In the results Section, we focus on five subregions of the OFC: namely, the lateral and medial parts of Brodmann area 11, Brodmann area 13, Brodmann area 14, and the subgenual part of Brodmann area 32. This parcellation is based on anatomical masks in standard MNI coordinates obtained from the BRAINNETOME atlas (https://atlas.brainnetome.org/, *Fan et al., 2016*). As can be seen in *Figure 11* below, these areas tile the entire OFC, except its most rostro-lateral part (which is Brodmann area 12). The standard MNI coordinates of each subregion barycenter are given in *Table 1* below.

**Table 1.** Anatomical coordinates of OFC subregions' barycenter.

| Anatomical region | Barycenter coordinates (left) | Barycenter coordinates (right) |
|---|---|---|
| Medial part of BA11 | 68,166,125 | 114,164,126 |
| Lateral part of BA11 | 81,146,127 | 100,149,127 |
| BA13 | 81,146,127 | 100,149,127 |
| BA14 | 84,182,114 | 97,176,114 |
| Subgenual part of BA32 | 87,167,110 | 96,169,101 |

To balance the statistical power across OFC subregions, we then removed the voxels that fall outside a 200-voxel sphere centered on the barycenter of the masks. This procedure yielded spherical ROIs with similar sizes across all ROIs.

FMRI data were preprocessed using SPM (https://www.fil.ion.ucl.ac.uk/spm/), following standard realignment and movement correction guidelines. Note that we excluded five participants from the narrow range group because the misalignment between functional and anatomical scans could not be corrected. In each ROI, we regressed trial-by-trial activations with SPM through a GLM that included one stick regressor for each trial (at the time of the gamble presentation onset), which was convolved with the canonical HRF. To account for variations in hemodynamic delays, we added the basis function set induced by the HRF temporal derivative. To correct for movement artifacts, we also included the six head movement regressors and their squared values as covariates of no interest. We then extracted the 256 trial-wise regression coefficients in each voxel of each ROI. Next, we removed potential between-session confounding effects by projecting the ensuing trial series onto the null space of a categorical session-encoding design matrix. This effectively provided a BOLD trial series $Y_{fMRI}$ that are deconvolved from the hemodynamic response function (**Dale, 1999**) and corrected for standard confounding effects. No spatial smoothing was applied to preserve information buried in spatial fMRI activity patterns. Finally, we concatenated the corrected multivariate fMRI activity patterns of left and right analogous ROIs, eventually yielding five OFC subregions.

## Model-free analysis of behavioral data

First, we describe peoples' behavior in terms of the probability of gambling given the gamble's expected value EV=0.5*(G-L), where G and L are the gamble's prospective gain and loss, respectively. For each participant, we binned trials according to deciles of EV, and measured the rate of gamble acceptance (**Figure 2**, upper-left panel).

Second, we regressed peoples' decision to gamble onto gains and losses. Within each participant, we fit the following logistic regression model: $p(g_i) = s(w_0 + w_G * G_i - w_L * L_i)$, where $g_i$ is the binary gamble decision at trial $i$, $G_i$ and $L_i$ are the prospective gain and loss of trial $i$, $w_0$ is the intercept or gambling bias, $w_G$ and $w_L$ are the sensitivity to gains and losses, respectively, and s(.) is the standard sigmoid mapping. Note that logistic model parameter estimates can be recombined to measure peoples' loss aversion ($\log(w_L/w_G)$). We then report within-subject parameter estimates at the group-level for random effect analyses (see **Figure 3** of the Results section). The logistic model can also be used to perform counterfactual model simulations. For each subject, we use the corresponding fitted parameters to evaluate the trial-by-trial probability of gamble acceptance that would have been observed, had this subject/model been exposed to the sequence of prospective gains and losses that each subject *of the other group* was exposed to. It turns out that such out-of-sample predictions of peoples' behaviour are (expectedly) inaccurate. More precisely, such logistic regression cannot predict the observed group-difference in peoples' gambling rate (see **Figure 4—figure supplement 1**).

Third, we performed a sliding window analysis: decisions were first partitioned into chunks of 16 consecutive trials each, which were then regressed against corresponding gains and losses using the same logistic model as above. From this, we obtain a set of logistic parameter estimates (intercept and sensitivity to gains and losses) per temporal window, per subject. Temporal changes in the ensuing loss aversion index can thus be followed as time unfolds (**Figure 2D**).

## RSA

Each ANN model of value synthesis makes specific trial-by-trial predictions of activity patterns within the integration layer that can be compared to multivariate fMRI signals in each OFC subregion. This enables us to evaluate the neurophysiological validity of candidate models. Here, we have chosen to rely on representational similarity analysis or RSA (*Diedrichsen and Kriegeskorte, 2017*; *Friston et al., 2019*; *Kriegeskorte et al., 2008*). In brief, RSA consists of evaluating the statistical resemblance between model-based and data-based *representational dissimilarity matrices* or RDMs, which we derive as follows. Let $Y$ be the $n_y \times n_t$ multivariate time series of (modeled or empirical) neural activity, where $n_y$ and $n_t$ are the number of units and trials, respectively. Note that, for model predictions, 'units' mean artificial elementary units in ANNs, whereas they mean voxels in a given ROI for fMRI data. First, we derive the $n_t \times n_t$ raw RDM $D_Y = \left\{ D_Y^{t,t'} \right\}$, where the matrix element $D_Y^{t,t'}$ measures the dissimilarity of neural patterns of activity between trial $t$ and trial $t'$: $D_Y^{t,t'} = 1 - corr\left(Y_t, Y_{t'}\right)$. By construction, these RDMs are invariant to affine transformations of activity patterns. In particular, this implies that the ensuing RSA are orthogonal to univariate analyses that rely on the mean activity within OFC subregions.

Second, we correct the raw RDM for autocorrelation confounds. To do this, we remove the average neural dissimilarity for each possible delay between trial pairs from the raw RDM. Note that this correction does not confound the existing relationship between neural dissimilarity and prospective gains and losses or EV, because these are randomized across trials.

When quantifying the neural encoding strength of prospective gains and losses, we simply regress the vectorized lower-left triangular part of $D_Y$ against Euclidean distances in either gains or losses concurrently (having included a constant term). This measures the gradient of neural dissimilarity per unit of gains and losses. We quantify the neural encoding strength of EV similarly (using a distinct regression analysis, to prevent regressor collinearities).

Finally, we measure the statistical similarity of $D_{Y_{ANN}}$ and $D_{Y_{fMRI}}$, where $D_{Y_{ANN}}$ is derived from activity patterns of the ANNs' integration layer and $D_{Y_{fMRI}}$ is derived from HRF-deconvolved multi-voxel fMRI trial series in each ROI, in terms of the Pearson correlation coefficient $\rho$ between the vectorized lower-left triangular part of $D_Y$. We then assess the group-level statistical significance of RDMs' correlations using one-sample t-tests on the group mean of Fisher-transformed RDM correlation coefficients $\rho$. Note that ANN-RSA summary statistics (such as RDM correlation coefficients) *do not* favor more complex ANNs (i.e. ANNs with more parameters, such as plastic ANNs). This is because, once fitted to behavioural data, ANNs produce activity patterns that have no degree of freedom whatsoever when they enter RDM derivations. In particular, this means that static ANNs can a priori show a greater RDM correlation than plastic ANNs. In turn, this enables a simple yet unbiased statistical procedure for comparing candidate ANN models. Importantly, this procedure is immune to arbitrary modeling choices such as the total number of units in ANN models.

# Additional information

### Funding

| Funder | Grant reference number | Author |
|---|---|---|
| Agence Nationale de la Recherche | ANR-20-CE37-0006 | Jean Daunizeau |

The funders had no role in study design, data collection and interpretation, or the decision to submit the work for publication.

### Author contributions

Jules Brochard, Resources, Data curation, Software, Formal analysis, Visualization, Methodology, Writing – original draft; Jean Daunizeau, Conceptualization, Software, Formal analysis, Supervision, Funding acquisition, Validation, Investigation, Methodology, Writing – original draft, Project administration

## Author ORCIDs
Jean Daunizeau https://orcid.org/0000-0001-9142-1270

## Decision letter and Author response
Decision letter https://doi.org/10.7554/eLife.80979.sa1
Author response https://doi.org/10.7554/eLife.80979.sa2

## Additional files

### Supplementary files
• MDAR checklist

### Data availability

All data analysed during this study are openly available from the https://openneuro.org/ website (https://doi.org/10.18112/openneuro.ds001734.v1.0.5). All the modelling and analysis code are available as part of the academic freeware VBA (https://github.com/MBB-team/VBA-toolbox/, *Rigoux et al., 2023*), which is under a GNU open-source license.

The following previously published dataset was used:

| Author(s) | Year | Dataset title | Dataset URL | Database and Identifier |
|---|---|---|---|---|
| Botvinik-Nezer R, Iwanir R, Poldrack RA, Schonberg T | 2020 | NARPS | https://doi.org/10.18112/openneuro.ds001734.v1.0.5 | OpenNeuro, 10.18112/openneuro.ds001734.v1.0.5 |

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

## Appendix 1

### Supplementary material

How efficient is efficient value synthesis

In what follows, we unpack the impact of efficient value synthesis using an exemplar numerical simulation of an ANN operating value synthesis (as described in Equations 3 to 6). For the sake of clarity, we will focus on a toy example made of 8 input units (4 for the gain sublayer and 4 for the loss sublayer) and 4 integration units. We assume that the network has been set to construct a readout value that is close to the objective gamble's EV, as defined by decision theory. We thus initialize the ANN parameters (including connectivity weights) using a random perturbation of ideal population codes, and then train the ANN to output EV over a unitary range of prospective gains and losses. .

As a reference point, we estimate the ensuing units' receptive fields, in terms of the units' response $z^{(k)}$ output to any $(G, L)$ pair of admissible prospective gains and losses. Note that we can associate a value to each point in that space ($EV = \frac{1}{2}G - \frac{1}{2}L$). This enables us to characterize the relationship between EV and each integration unit in terms of their average response to the subset of pairs of prospective gain and loss that lie along iso-value lines. Note that this relationship is statistical in essence (as opposed to causal), since units respond to prospective gains and losses (as opposed to EV). Nevertheless, this relationship may be strong (in a statistical sense) because the network has been trained to signal value (which is constructed as a weighted sum of integration units' responses). *Figure 1—figure supplement 1* below summarizes this analysis.

One can see that each integration unit has a receptive field that spans a specific range of prospective gains and losses. In turn, it exhibits an idiosyncratic statistical relationship to EV. Note that these relationships are slightly ambiguous (variations of response outputs within EV bins). This is because units typically respond nonlinearly to prospective gains and losses. Nevertheless, despite those multiple nonlinearities, the ANN's readout value profile is clearly linear in the prospective gains and losses that compose gambles. *Finally* the ANN readout value's sensitivities to prospective gains and losses match their theoretical values (namely: 0.5), which simply means that the network has been correctly trained. .

We then simulate self-organized plasticity according to *Equation 17* over two restricted domains of prospective gains and losses: (i) the 'narrow' domain is symmetrical and such that the spanned range of gains is the same as that of losses, and (ii) the 'wide' domain is asymmetrical and such that the spanned range of gains is twice that of losses. Finally, we quantify the efficiency of value synthesis, both before and after self-organized plasticity has reshaped the system's connectivity, in terms of the expected log steepness of units' activation functions (*Equation 9*) and in terms of the system's resilience to neural noise. We measure the latter by reading out the noisy value response of the network to gain/loss pairs spanning the corresponding domain, having added Gaussian neural noise to integration units' activity patterns with variances ranging from 1/256 to 2 (*Equation 8*), and quantifying the rate of 'stable choices,' i.e., gamble decisions that are identical to those taken without neural noise.

The results of this analysis are summarized in *Figure 1—figure supplement 2* below.

One can see that self-organized plasticity has increased the average steepness of units' activation functions. This is reassuring, since it was derived to operate a gradient ascent on this metric. More importantly, self-organized plasticity tends to increase the system's resilience to neural noise on integration units.

### What does self-organized plasticity do to the network?

To address this question, we first reproduce the analysis of *Figure 1—figure supplement 1*, this time after self-organized plasticity has modified the network connectivity, while being exposed to the 'wide.' domain above (see *Figure 1—figure supplement 3:* below).

Comparing *Figure 1—figure supplement 1* and *Figure 1—figure supplement 3:* shows that the receptive fields of integration units have been altered, even beyond the domain of prospective gains and losses that were spanned during efficient integration (except, maybe for unit #3, which underwent very small changes in its connectivity to the feature layer of the network). In particular, units #1 and #2 are now mostly sensitive to losses, whereas unit #4 has become mostly sensitive to gains. In turn, this eventually distorted the readout value profile. Importantly, the readout value profile now shows, within the spanned domain of prospective gains and losses, distorted sensitivities

to gains and losses. More precisely, the sensitivity to losses $\omega_L$ is now about twice that of gains $\omega_G$. Self-organized plasticity over this domain of prospective gains and losses would thus eventually yield loss aversion.

One can also see that self-organized plasticity has changed the statistical relationship between integration units' responses and gamble EVs. In brief, all integration units now show stronger response variations across the spanned range of EVs, i.e., inputs to integration units now tend to span the non-saturating range of their activation function (this is what is measured in the left panel of *Figure 1—figure supplement 2*). In addition, this relationship also tends to become more ambiguous, i.e., there are stronger variations of response outputs within EV bins. This is due to the induced distortion of the units' receptive fields. Nevertheless, self-organized plasticity results in an apparent phenomenon of (partial) adaptation to the range of EVs. Practically speaking, should one attempt to detect those integration units that show a significant relationship with EV, one would conclude that self-organized plasticity seems to have 'recruited' integration units (that would otherwise show no strong covariation with EV). But here again, this phenomenon is only apparent, because EV is not an input to the network.

Now, we ask whether and how self-organized plasticity changed the information content within the integration layer. To do this, we rely upon RSA, which can be reproduced in an empirical setting where neural populations are sampled over an arbitrary domain of prospective gains and losses. We first measure the neural dissimilarity $d(i,j,k,l)$ of each possible pairwise combination of gains and losses, in terms of the correlation (across integration units) between the corresponding response patterns, i.e.,: $d(i,j,k,l) = 1 - corr[z(G_i, L_j), z(G_k, L_l)]$. By construction, this measure of dissimilarity is bounded between 0 (when the response patterns are colinear) and 2 (when they are anti-correlated). We then construct the RDM for EV as follows. Recall that each pair of prospective gain and loss belongs to a given EV bin. The RDM element that corresponds to a pair $\{EV_m, EV_n\}$ of EV bins is estimated as the average dissimilarity $\langle d(i,j,k,l) \rangle$ over all combinations for which $(G_i - L_j)/2 \in EV_m$ and $(G_k - L_l)/2 \in EV_n$. We also construct RDMs for prospective gains and losses similarly. If the network contains information about a given variable, then the corresponding RDM should show a diagonal pattern, such that the neural dissimilarity increases with the absolute difference between elements of the pair. Accordingly, we define the neural encoding strength of EV (resp., prospective gains or losses) in terms of the gradient of neural dissimilarity per unit of absolute difference of EV (resp., prospective gains or losses). This analysis can be reproduced after self-organized plasticity has modified the integration layer's receptive fields, while exposing the ANN to the wide (asymmetrical) domain above. *Figure 1—figure supplement 4* below summarizes the results of these analyses.

In brief, self-organized plasticity strongly strengthened the encoding of all variables, i.e., the ANN's integration layer contains more information about EV and prospective gains/losses than before self-organized plasticity. Interestingly, although this network has been trained to perform value synthesis, the encoding of EVs after self-organized plasticity is relatively weaker than before (smaller variations of dissimilarity along directions orthogonal to iso-EV lines), at least when compared to prospective gains and losses. This is because changes in integration units' receptive fields eventually reduced the redundancy between integration units, which now tend to decompose the array of feature units' outputs into their independent sources of variations (here: prospective gains and losses). Nevertheless, as one can check in *Figure 1—figure supplement 4*, applying the readout weights on the integration units' response still yields a reasonable readout value profile (that exhibits loss aversion).

## Logistic regression of behavioral data: Postdiction and out-of-sample predictions

The logistic regression analysis of peoples' sensitivity to gains and losses provides a trial-by-trial prediction of gamble acceptance, for each pair of prospective gain and loss. But are hard decisions (i.e. when EV is close to zero) as well explained as easy decisions (e.g., when there is a strong incentive to gamble)? To address this question, we binned trials according to EV deciles (see Methods), and measured the rate of the logistic model's postdiction error (see *Figure 4—figure supplement 1A*).

Note: hereafter, we refer to 'postdiction"' as model predictions on data that was used to fit the model's parameters. In contrast, 'out-of-sample predictions' are model-proper predictions on yet unseen data.

One can see that the logistic regression model achieves relatively similar postdiction error profiles in both groups. In particular, easy decisions (either low or high EV) exhibit much lower postdiction errors than difficult decisions (EV around zero).

We also performed counterfactual model simulations: for each subject, we simulated the trial-by-trial gamble acceptances that would have been observed, under the logistic model, had this subject/model been exposed to the sequence of prospective gains and losses that each subject of the other group was exposed to (see *Figure 4—figure supplement 1B*). Those out-of-sample predictions are inaccurate: within the EV range that are common to both groups, it wrongly predicts that the gambling rate should be higher in the wide-range group than in the narrow-range group. This is not surprising, because predictions of the logistic model are entirely determined by the pair of prospective gain and loss (and cannot exhibit contextual effects). In other words, for a given gamble (defined in terms of its constituent gain and loss), the logistic model makes an out-of-sample prediction that corresponds to its equivalent postdiction (or is an extrapolation of it, if performed outside the EV range it was originally trained with).

Finally, we asked whether inter-individual differences in gambling rate (within the common EV range) were better explained in terms of inter-individual differences in loss aversion (i.e. the ratio $w_L/w_G$, where $w_L$ and $w_G$ are estimates of behavioral sensitivity to gains and losses, respectively) or in terms of gambling bias ($w_0$). This is summarized in *Figure 4—figure supplement 1C, D*. In brief, loss aversion negatively correlates with gambling rate within the common EV range (wide gain range group: $r=0.59$, $p<10^{-3}$, narrow gain range group: $r=0.26$, $p=0.067$, both groups together: $r=0.53$, $p<10^{-3}$).

## Representational dissimilarity matrices within OFC subregions

In the main text, we summarize the information content within OFC subregions in terms of the gradients of neural dissimilarity per unit of prospective gains, losses, or EV. For the sake of completeness, we report below the underlying representational dissimilarity matrices, where trials have been binned according to prospective gains (gain-RDMs, *Figure 5—figure supplement 1*), losses (loss-RDMs, *Figure 6—figure supplement 1*), or EV (EV-RDMs, *Figure 7—figure supplement 1*).

Note that a strong neural encoding of gains would correspond to strong variations in neural dissimilarity along directions orthogonal to iso-distance lines. Such RDMs would show a clear bandwise-diagonal structure, where neural dissimilarity would increase when moving away from the main diagonal. One can see that gain-RDMs within the lateral part of BA11 are the closest to this ideal situation.

Here again, loss-RDMs within the lateral part of BA11 show the strongest encoding of prospective losses. Interestingly, the loss-RDM of BA11 for the narrow gain range group seems to exhibit a block structure, such that trials seem to be partitioned into two classes according to whether they lie either below or above the median loss. This suggests that the encoding of losses is not, strictly speaking, linear. Nevertheless, this does not confound the analysis of encoding strength which we report in the main text.

Except in the lateral part of BA11, all EV-RDMs exhibit antidiagonal elements with weak neural dissimilarity. This means that multivariate fMRI patterns of trials that correspond to either very low or very high EV are similar to each other. This is the reason why the average neural dissimilarity tends to eventually *decrease* for extreme absolute EV distances (*Figure 7* of the main text).

## Univariate analysis of fMRI data

In the main text, we report the results of multivariate (RSA) analyses of fMRI data, in five distinct subregions of the orbitofrontal cortex. For completeness, we also performed univariate analyses, having summarized the activity within OFC subregions in terms of the average trial-by-trial BOLD response over voxels (see Methods section for parametric estimation of trial-by-trial BOLD responses), and corrected the ensuing univariate responses for between-session confounding effects.

Here, we rely on two distinct general linear models or GLMs (*Friston et al., 1995*). All models incorporated only one event per trial, which was the onset of the gamble presentation, convolved with a canonical hemodynamic response function, as well as with its temporal derivative (*Hopfinger et al., 2000*). In the first GLM, we used two parametric modulations of the trial epoch regressor: namely prospective gains (G) and losses (L). We regressed the average trial-by-trial BOLD response

against prospective gains and losses concurrently, using a GLM including an offset (constant term). The ensuing GLM parameter estimates measure the within-subjects gradient of BOLD responses per unit of prospective gains and losses. In the second GLM, we used only one parametric modulation, i.e. the gamble's expected value EV=0.5*(G-L). Here, the ensuing GLM parameter estimates measure the within-subject's gradient of BOLD response per unit of EV. Note that gain, loss and EV regressors were mean-centered but not rescaled to allow for a proper between-group comparison of neural sensitivity to these factors. For both regression analyses, we reported the ensuing parameter estimates at the group level and performed within-group and between-group statistical significance tests using standard one-sample and two-sample F-tests, respectively. The results of these analyses are summarized in *Figure 7—figure supplement 2* below.

One can see that EV is significantly encoded in both groups of subjects in subregions BA14 and BA32. At the very least, the latter result survives the correction for multiple comparisons across OFC subregions. This reproduces established univariate results regarding the encoding of value in the fMRI literature. However, although some regions also show a significant encoding of prospective losses, nowhere in the OFC is there a significant encoding of both gains and losses, in both groups of subjects. This is strikingly different from the multivariate fMRI analyses results summarized in *Figures 4–6* of the main text, which clearly exhibit stronger statistical power. We note that, by construction, these two analyses are orthogonal to each other. This is because we chose to measure neural dissimilarity in terms of Pearson correlations across voxels, which are invariant under isotropic (within OFC subregions) affine transformations of voxel-wise trial series.

## Bayesian priors on ANNs' parameters

We fit each candidate ANNs to observed trial-by-trial gamble decision sequences using a dedicated Bayesian approach, which requires setting specific prior distributions on model parameters. These priors distributions are summarized on *Appendix 1—table 1* below.

**Appendix 1—table 1.** Parameters' priors for biologically constrained artificial neural networks (ANNs).

| Parameter | Distributions | Rational |
|---|---|---|
| Firing rate threshold | $\mu \sim \mathcal{N}\left(\frac{j}{n+1}, 1\right)$ | Regular tiling of inputs |
| Activation function slope | $\sigma = e^{\theta}$ with $\theta \sim \mathcal{N}(0,1)$ | Partially overlapping tiling of inputs |
| Initial connectivity | $c^{(i,j,k)} \sim \mathcal{N}\left(sign\left(i - 1/2\right), 1\right)$ | EV population code |
| Value readout weights | $w^{(k)} \sim \mathcal{N}(1,1)$ | EV population code |
| Plasticity magnitude | $\alpha = e^{\theta}$ with $\theta \sim \mathcal{N}(0,1)$ | Non informative prior |
| Plasticity rate | $\beta = \frac{1}{1+e^{-\theta}}$ with $\theta \sim \mathcal{N}(0,1)$ | Non informative prior |

All parameter notations are defined in the Methods section of the main text. Note that we performed all the analyses using the VBA academic freeware (*Daunizeau et al., 2014*). Although this toolbox only handles Gaussian prior distributions, native (Gaussian) VBA parameters can be passed through arbitrary mappings prior to entering model computations. This enables VBA to enforce any required constraint (see e.g. *Daunizeau, 2017*). This is the case here for the slopes of activation functions, as well as for the self-organized plasticity magnitude and rate parameters. In *Appendix 1—table 1*, θ denotes VBA native parameters: they are given Gaussian prior distributions, and then passed through the appropriate nonlinear mapping to enforce positivity or bounding constraints.

In addition to the prior distributions for ANN parameters given in *Appendix 1—table 1* above, we set the number of attribute-specific and attribute-integration units to $n_x = 4$ (per sublayer) and $n_z = 8$, respectively. We also rescaled the ANN's inputs (i.e. prospective gains and losses) such that they lie within the unit interval. To ensure an unbiased comparison between decision attributes and/or between groups of participants, this rescaling is invariant across decision attributes and groups, and such that the upper bound of the unit interval corresponds to 150% of the maximum prospective

gain or loss (here: 40$). This also prevents artifactual ceiling effects in the population code of the attribute layer (whose units with the highest firing rate thresholds always remain weakly activated).

## ANNs' behavioral postdiction and prediction accuracy

We fit the static and plastic ANN models to each subject's sequence of gamble decisions. In what follows, we provide summary statistics of our ANN-based behavioral data analyses.

In addition to the balanced accuracy scores given in the main text, *Appendix 1—table 2* below gives the group average percentage of explained behavioral variance ($R^2$) and its standard deviation (across participants) for each model (including the logistic regression model, for comparison purposes), and each group.

**Appendix 1—table 2.** Mean $R^2$ and its standard deviation for each model, for both groups.

| | Narrow range group | | Wide range group | |
|---|---|---|---|---|
| | Mean | Std | Mean | Std |
| Logistic | 0.74 | 0.02 | 0.66 | 0.02 |
| Static ANN | 0.73 | 0.02 | 0.62 | 0.03 |
| Plastic ANN | 0.75 | 0.02 | 0.66 | 0.02 |

In brief, all models achieve similar fit accuracies, in both groups, and no pairwise comparison between models reaches statistical significance.

But do the models yield similar types of postdiction errors? To address this question, we bin trials according to EV deciles, prior to averaging the rate of postdiction error across subjects. *Figure 8—figure supplement 1* below summarizes the accuracy of behavioral postdiction as a function of gambles' expected value, for all ANNs.

As was the case for the logistic regression model, ANNs exhibit high fit accuracy for easy decisions (extreme EVs) and lower fit accuracy for hard decisions (EV around 0).

## Parameter estimates of plastic ANNs

Of particular interest in our ANN-based analysis of behavioral data are the parameters that control the self-organized plasticity rule of efficient value synthesis (plasticity magnitude α and rate β parameters). *Figure 8—figure supplement 2* below shows the empirical histograms of parameter estimates, for both groups of participants.

Overall, empirical distributions of parameter estimates are qualitatively similar in both groups of participants. When comparing groups of participants with respect to either α or β parameter estimates, nothing reaches statistical significance.

We also asked whether fitted plasticity parameters explained inter-individual differences in observed loss aversion, as measured using the logistic model (see *Figure 8—figure supplement 2C, D*). Nothing really stands out for plasticity rates, but the situation is quite different for plasticity magnitudes. Within the wide gain range group, there is a significant correlation between plasticity magnitudes and loss aversion across subjects ($r=0.47$, $p<10^{-3}$), while this correlation is negative within the narrow gain range group ($r=-0.32$, $p=0.024$). That the sign of this correlation is different in both groups makes sense, given that the effect of self-organized plasticity is to decrease loss aversion within the narrow gain range group, whereas, if anything, it tends to increase it within the wide range group (see *Figure 4D* in the main text).

## On the diversity of response profiles in ANNs' integration units

OFC neurons are notoriously diverse in their response profile, but a consistent finding is that, in the context of value-based decision-making, they can be classified in terms of so-called 'choice cells,' 'chosen value cells', and 'offer value cells' (*Padoa-Schioppa and Assad, 2006*; *Padoa-Schioppa and Assad, 2008*). Given that this can be considered a pre-requisite for any computational model of value integration in the OFC, we asked whether ANNs reproduce this known property of OFC neurons.

For each subject, we thus tested whether the response of integration units correlates (across trials) with choice, chosen value, and/or gamble value, where value is defined as the weighted sum of gains and losses (according to the static logistic model parameter estimates). Integration units are

then classified in terms of which variable it correlate most with (but none of these labels is assigned if the best correlation is not significant). In brief, 'choice units' show response outputs that vary in a quasi-categorical manner with value (i.e. that discriminate trials in which the value of gambling is either positive or negative), 'chosen value units' exhibit a ReLU-like relationship with value (null when the value of gambling is negative, and increasing with value when it is positive), and 'offer value units' show a quasi-linear relationship with value. The results of this analysis are summarized in *Figure 9—figure supplement 1* below.

In previous electrophysiological experiments, the detection rate of 'offer value,' 'chosen value,' and 'choice' cells within OFC neurons depends upon the delay to stimulus onset. This is because the detection analysis is typically performed at each time point within a given peristimulus time window. We thus extracted the frequency of 'offer value,' 'chosen value,' and 'choice' cells detected at OFC neurons' response peak, i.e., about 300 msec after stimulus onset (see Figure 4 in *Padoa-Schioppa and Assad, 2006*). For comparison purposes, we then normalized these estimated frequencies to remove non-responsive OFC neurons (see horizontal grey bars in *Figure 9—figure supplement 1*). We find that, although HP-ANN's integration units were not at all designed to encode these quantities, they eventually reproduce the known response variability observed in OFC neurons (although few units are eventually classified as 'choice cells' in the narrow gain range group). This means that our ANN models reproduce the known diversity of response profiles within OFC neurons. Importantly, the response profiles are almost identical for both groups. In particular, we find that about 34% of integration units are classified as 'chosen value' cells. This is interesting, because the computational role of these units is in fact exactly the same as that of units that are classified as 'offer value' cells: together, they form a population code for the subjective value of gambling. In other words, this classification is not directly relevant for guessing the underlying computational role of integration units.

## Value synthesis under efficient coding of gains and losses

In the main text, we assumed that neural noise would be acting on the output of integration units. But it may also be acting on its inputs, or equivalently on the outputs of the attribute units:

$$\begin{cases} \upsilon_t^{(k)} = \sum_{i=1}^{n_u} \sum_{j=1}^{n_x} C^{(i,j,k)} \widetilde{x}_t^{(i,j)} \\ \widetilde{x}_t^{(i,j)} = x_t^{(i,j)} \left( u_t^{(i)} \right) + \eta_t^{(i,j)} \end{cases} \tag{A1}$$

where $\eta_t^{(i,j)}$ is some (uncontrollable) neural noise that competes with the 'utile' component $x_t^{(i,j)} \left( u_t^{(i)} \right)$ of the responses of attributes units.

Similarly to *Equation 9*, this also induces an information loss $IL$:

$$IL = -MI \left( \widetilde{x}, x \right) \underset{\eta \to 0}{\to} K - H\left[u\right] - \sum_i \sum_j E \left[ ln \left| \frac{\partial f_x^{(i,j)}}{\partial u^{(i)}} \right| \right] \tag{A2}$$

In our framework, we do not consider how brain systems upstream of the attribute layers extract gain and loss information from visual stimuli and project it onto attribute layers. Rather, we assume that prospective gains and losses are encoded into population codes within attribute layers. Nevertheless, we can still model efficient coding at the level of attribute units. More precisely, efficient coding of gains and losses can then be achieved by modifying the response properties of attribute units to decrease the information loss $IL$ in *Equation A2* i.e.,:

$$\Delta \theta^{(i,j)} = -\alpha \frac{\partial IL}{\partial \theta^{(i,j)}} = \alpha \frac{\partial}{\partial \theta^{(i,j)}} E \left[ ln \left| \frac{\partial f_x^{(i,j)}}{\partial u^{(i)}} \right| \right] \tag{A3}$$

where $\theta^{(i,j)}$ is the $2 \times 1$ vector of location and scale parameters of attribute units' activation functions.

This is but a proxy for the impact of self-organized plasticity upstream attribute layers. Note that there is no nonlocal term in **Equation A3**, because the entropy term $H[u]$ is beyond the network's control.

Let $\Delta\theta_t^{(i,j)}$ be the change of location and scale parameters at trial or time $t$. Similarly to efficient value synthesis, an online implementation of **Equation A3** is operated as follows:

$$\Delta\theta_t^{(i,j)} \approx \alpha\left(1-\beta\right)\Delta\theta_{t-1}^{(i,j)} + \alpha\beta\frac{\partial}{\partial\theta^{(i,j)}}ln\left|\frac{\partial f_x^{(i,j)}}{\partial u_t^{(i)}}\right| \tag{A4}$$

where $\beta$ (note: $0 < \beta < 1$) controls the exponential decay of past samples' weights in the moving average operator (**Equations 13-14** in the main text).

If the ANN is equipped with sigmoid activation functions, then simple analytical derivations show that **Equation A4** reduces to the following update rule for location and scale parameters of attribute units:

$$\begin{cases} \Delta\mu_{x,t}^{(i,j)} = \alpha\left(1-\beta\right)\Delta\mu_{x,t}^{(i,j)} - \alpha\frac{\beta}{\sigma_{x,t}^{(i,j)}}\left(1 - 2x_t^{(i,j)}\right) \\ \Delta\sigma_{x,t}^{(i,j)} = \alpha\left(1-\beta\right)\Delta\sigma_{x,t}^{(i,j)} - \alpha\frac{\beta}{2\sigma_{x,t}^{(i,j)2}}\left(1 - 2x_t^{(i,j)}\right)\left(u_t^{(i)} - \mu_{x,t}^{(i,j)}\right) \end{cases} \tag{A5}$$

**Equation A5** is the equivalent of **Equation 2** of the main text, i.e., it describes how the properties of attribute units within the network should modify their response properties to operate efficient coding of attribute inputs. In what follows, we simply refer to **Equation A5** as *efficient coding* of attributes. Note that **Equation A5** would also modify the receptive fields of integration units within the ANN. In principle, it could thus yield effects that are qualitatively similar to those of efficient value integration.

But what are the neural and behavioral impacts of this efficient coding mechanism? To address this question, we reproduced the same analyses as for efficient value synthesis.

First, we asked whether **Equation A5** would produce apparent value range adaptation (**Figure 2** in the main text) in integration units. We randomized the trained connections of ANNs that operate value synthesis prior to exposing them to four different series of 256 decision trials made of prospective gains and losses with a predefined range. As before, we considered two ranges (either narrow or wide) for both prospective gains and losses, and exposed the ANNs to each of the 2 × 2 range combinations. We then averaged the activity of integration units, after having binned trials into EV deciles. We repeated this procedure 1000 times, and **Figure 10** in the main text summarizes the results of this analysis. One can see that the lower and upper limits of units' mean responses are tied to the bounds of the spanned EV range, as is the case for static value synthesis (see **Figure 2C and D** in the main text). In turn, the slope of the relationship between EV and integration units' mean responses does not seem to vary strongly with the range of spanned EVs. In other words, the efficient coding mechanism in **Equation A5** does not induce apparent value range adaptation in the univariate response of integration units.

Second, we asked whether and how the shape of spanned the gain/loss domain modifies the neural and behavioral sensitivities to prospective gains and losses. We thus performed the same series of Monte-Carlo simulations as for efficient value synthesis. We simulated the response of the ANN, modified according to **Equation A5** to operate efficient coding of attributes, to a series of 256 gambles, after having randomized the trained connectivity within the network. We systematically varied the spanned range of prospective gains and losses, and measured the behavioral sensitivities to gains and losses (in terms of the gradient of the readout value per unit of gain or loss) and the neural sensitivity to EV (in terms of the gradient of the integration layer's neural dissimilarity per unit of EV). We repeated this procedure 1000 times, and **Figure 10—figure supplement 1** below summarizes the results of this analysis.

In brief, the behavioral impact of efficient coding is qualitatively similar to that of efficient value synthesis. More precisely, the behavioral sensitivity to gains (resp., losses) decreases as the spanned range of gains (resp., losses) increases (**Figure 10—figure supplement 1D, E**). Note that cross-attribute spillover effects are also present here, and to a greater extent than under efficient

value synthesis. In addition, efficient coding of attributes induces a similar effect on behavioral loss aversion, which follows the ratio of the spanned range of gains relative to the spanned range of losses (*Figure 10—figure supplement 1F*). This implies that behavioral observations alone would not disambiguate efficient value synthesis (as described in *Equation 2* in the main text) from efficient coding of attributes (as described in *Equation A5*).

Also, the neural encoding strength of EV in the integration layer is similarly impacted by the shape of the spanned gain/loss domain. That is, the neural encoding strength of EV decreases when the spanned range of either gains or losses increases. Thus, this effect does not disambiguate efficient value synthesis from efficient value coding of attributes. However, the neural sensitivity to gains and losses does not react to the shape of the spanned gain/loss domain as they do under the efficient value synthesis scenario (see *Figure 10C, D* in the main text). The difference here is twofold. First, the cross-attribute spillover effect dominates, i.e., the most salient effect is that the neural sensitivity to gains (resp. losses) increases when the spanned range of *losses* (resp., gains) increases. Second, if anything, the within-attribute effect seems to be opposite to that of efficient value synthesis. That is, the neural sensitivity to gains (resp. losses) *increases* when the spanned range of gains (resp., losses) increases. At the very least, this holds for small to intermediate ranges of gains and losses; this effect reverses for extreme ranges of gains and losses (most likely because of artefactual ceiling effects). This is interesting, because this is clearly at odds with the neural predictions of the efficient value synthesis scenario. Importantly, this directly contradicts the fMRI data sampled in the lateral part of Brodman area 11 (see *Figure 5* in the main text).

We then reproduced the same model-based analyses as with the efficient value synthesis scenario.

First, we fit the ANN model equipped with efficient coding of attributes to each participant's trial-by-trial gamble decisions, and extract counterfactual out-of-sample behavioral predictions when exposing the fitted ANNs to the gamble series of the other group. We also performed the sliding window analysis to investigate the temporal dynamics of loss aversion, as captured by the scenario of efficient coding of attributes. The results of these analyses are summarized in *Figure 10—figure supplement 2* below.

In brief, behavioral postdictions under the scenario of efficient coding of attributes are as accurate as under the efficient value synthesis scenario. In particular, the postdicted dynamics of loss aversion exhibit the same qualitative properties (compared with *Figure 7C*). However, out-of-sample behavioral predictions are not as convincing: although the model does predict a behavioral change, predicted gamble rates are still higher for the wide gain range group (mean gambling rate=0.58±0.02) than for the narrow gain range group (mean gambling rate=0.48±0.02). This observation is partially confirmed when comparing the absolute out-of-sample prediction error of both plastic ANN models. In brief, we found that out-of-sample behavioral predictions under efficient coding of attributes were significantly less accurate than under efficient value synthesis for the wide range group (p=0.006, *F*=12.8), but not for the narrow range group (p=0.31, *F*=1.15).

Second, we reproduced the same RSA analyses as before, i.e., we evaluated the similarity between the ANN model and fMRI data sampled in the same five OFC subregions. The results of these analyses are summarized in *Figure 10—figure supplement 3* below.

No OFC subregion reaches statistical significance in both groups, for all types of RDMs. Importantly, this also holds for the lateral part of Brodman area 11 (only 5 out of 8 tests are significant). We note that, irrespective of the type of RDM considered, nowhere in the OFC is the comparison between both plastic models statistically significant.

