## [Editor Report]

This valuable manuscript proposes a neural network mechanism for range adaptation for value-based decision making. The authors present solid evidence for the proposed mechanism.

---

## [Decision Letter]

**Decision letter after peer review:**

Thank you for submitting your article "Synaptic plasticity in the orbitofrontal cortex explains how risk attitude adapts to the range of risk prospects" for consideration by *eLife*. Your article has been reviewed by 3 peer reviewers, one of whom is a member of our Board of Reviewing Editors, and the evaluation has been overseen by Michael Frank as the Senior Editor. The reviewers have opted to remain anonymous.

Essential revisions:

1) As you can see in the individual comments, all reviewers thought that your paper addresses an important topic. However, a central concern raised by all reviewers was that a substantial part of the model performance is driven by the activation function – yet, throughout the manuscript, you mainly discuss the role of Hebbian plasticity and largely ignore the effects of the activation function. There was agreement among reviewers that this is not warranted and that your manuscript requires substantial reframing, clarification, justification, and discussion. Reviewers would expect an adequately revised manuscript to look quite different from the current version, with an equal focus on all factors that allow the model to account for the observed changes across different ranges.

2) There are additional comments in the individual critiques that would be important to address, specifically regarding aspects of the analysis and interpretation.

*Reviewer #1 (Recommendations for the authors):*

1) Please explain on page 4 of the introduction why Hebbian plasticity should lead to spill-over effects. This is not very intuitive.

2) Interpretation of the null findings in the fMRI data (page 4 of results) is problematic because it is unclear whether they reflect a true null effect or a lack of sensitivity. Although this is true for all null results, it is particularly problematic for re-analyses, as the study was not designed or powered to test this question. It would be best to remove these results from the paper.

3) There is a fundamental difference between gaussian and sigmoidal activation functions. It would be important to include an adequate discussion of the assumptions and implications of these different functions in the main text.

4) The out-of-range predictions of the best model (Figure 3, lower-right panel, HP-ANN (gauss)) are not very convincing when considering the entire EV range. Model performance should be compared for the entire EV range, not just the common range. What does this mean for the proposed mechanism?

5) The paper focuses on Hebbian plasticity as a mechanism for context effects but judging from Figure 3, the choice of activation function has a comparable effect. Indeed, HP does almost nothing for models with sigmoidal activation functions, and HP only improves the out-of-range prediction for the common EV range but does very little for the uncommon range. A more balanced presentation that also discusses the type of activation function as a mechanism of adaptation would be important.

6) Are the RSA results in Figure 4 based on the ANNs with gaussian activation function? It would be important to show RSA results for all 4 ANNs in Figure 4, so it is possible to compare the results across models. Also, please add labels to the plot axes.

7) The proportion of offer and chosen value neurons shown in Figure 6 is opposite to what has been reported in the OFC of non-human primates. It would be good to discuss this discrepancy. Also, are the same proportions found for all 4 ANNs?

8) Figure 4: the spheres shown for BA11 are in the posterior medial rather than the lateral OFC. Please double-check the anatomical location of these ROIs. Are they really in the lateral OFC? Also, it would be good to provide center coordinates for the ROIs. In general, it would be important to better describe how the ROIs were generated. What were the search terms used in NeuroQuery? Also, NeuroQuery generates meta-analytic activation maps, not maps of anatomical structures. It would be better to use actual anatomical ROIs for fMRI data analysis.

*Reviewer #2 (Recommendations for the authors):*

1) Range adaptation is not shown directly in ANN units. Specific questions:

a) How do ANN units respond across different input values? How variable are these response patterns across units?

b) How do these response patterns vary with range?

c) Do these patterns (at the individual unit or mean level) resemble neuronal data from the orbitofrontal cortex (OFC)? Should it be expected to?

d) The example units that are shown (Figure 7) seem potentially different from previously reported neuronal data from OFC. For example, the dynamic range of the model covers only a narrow subset of the input space and varies substantially across conditions, whereas firing rates in OFC neurons tend to span the full range of possible values, and firing rates change only slightly between conditions [ref,ref]. Is this discrepancy just a side effect of showing results in terms of loss units rather than expected value? How should we interpret this apparent discrepancy?

e) ANN unit responses are compared to neuron classes observed in OFC (Figure 6). What does the mean ANN unit in each category look like, and how does this compare to the OFC responses referenced?

2) Modeling results focus on Hebbian networks with gaussian activation functions. Specific questions:

a) Is there a physiological motivation for the model, or is this primarily for mathematical convenience?

b) Do the main qualitative results (range effect on risk sensitivity) require a non-monotonic activation function?

c) How do the properties of the response functions (saturation and (non)-monotonicity) affect responses of ANN units?

d) There are visible differences between model behavior for Hebbian networks with sigmoidal and gaussian activation functions. How should these differences be interpreted? Does this lead to any predictions or constraints on what physiological implementation is consistent with this algorithm?

3) Out-of-sample predicted choices in the Hebbian model with gaussian activation function seem unintuitive for more extreme parts of the value range. E.g. for the out-of-sample wide range predictions, the model seems to over-predict risk aversion to the point that choice probabilities saturate and ~0.6 for increasingly high-value options. For the narrow range, out-of-sample prediction behavior even appears to be non-monotonic, leading to increased acceptance of extremely high loss options.

a) How should this be interpreted?

b) Does this depend on model parameters like update rate or covariance threshold?

c) In these parts of the range, how does the Hebbian model compare to alternate models, such as the other ANNs or logistic regression?

4) The relationship between neural responses and ANN activity relies on representational similarity analysis (RSA). However, significant RDM correlations could arise as a byproduct of the fact that both ANN output and BOLD activity in select regions correlate with behavioral choice patterns. Is there evidence that the correlation between Hebbian ANNs and BOLD activation reflects more than average choice patterns?

5) The authors make the strong claim that this model is the mechanistic explanation for adaptation in orbitofrontal cortex, but there is little comparison with previous models. Divisive normalization and other forms of adaptation to the value range are discarded based on a qualitative argument from the behavioral data. However, given that the Hebbian ANNs also produce some counterintuitive behavioral predictions, it is not obvious that they are better at accounting for observed patterns in neuronal adaptation or behavior. Addressing the following questions could clarify whether there is an argument for Hebbian ANNs over alternate mechanisms of adaptation:

a) How do behavioral predictions and RSA results from HP-ANNs compare quantitatively to results from other models of adaptation, including models of adaptation at the input stage?

b) Can Hebbian ANNs account for other previously observed patterns of behavior across different value ranges, such as stability of relative values across ranges in two-option choices and range-dependent decoy effects []?

c) Are there specific predictions that arise from Hebbian networks that could be tested in later work and used to differentiate between competing models?

*Reviewer #3 (Recommendations for the authors):*

1) The paper assumes loss aversion as the primary behavioral factor, which is fine. However, it may be worth briefly mentioning the limitation that with the present experimental design it is impossible to dissociate loss aversion from risk aversion (see e.g. Williams et al., 2021, PNAS).

[Editors' note: further revisions were suggested prior to acceptance, as described below.]

Thank you for resubmitting your work entitled "Efficient value synthesis in the orbitofrontal cortex: how does peoples' risk attitude adapts to the range of risk prospects?" for further consideration by *eLife*. Your revised article has been evaluated by Michael Frank (Senior Editor) and a Reviewing Editor.

All reviewers agreed that the revised manuscript has been improved. However, given the revised manuscript is essentially a fundamentally new manuscript, there is a new set of comments that would need to be addressed, as outlined below.

*Reviewer #1 (Recommendations for the authors):*

I appreciate the authors' effort and dedication in re-working the manuscript. The revised manuscript is a fundamentally new and improved paper with new conclusions. I believe it could make an important contribution to the field and I remain enthusiastic. However, given most of the manuscript has changed, I have new comments that I think should be addressed

1) In general, the manuscript is quite long with extensive supplementary analyses. I believe the paper could be streamlined by highlighting the most important aspects and reducing the extent to which details are discussed in the main text.

2) All relevant information to understand the plots should be embedded within the figure rather than just the figure legends. It is cumbersome for readers to constantly have to consult the legends to understand what is shown in the figure. For instance, there is no label for the different colored plots (red/back, red/blue) or line styles in ANY of the figures. Also, some legends (Figure 4) refer to a color code, but this code is not provided. Moreover, there are no axis ticks and/or axis tick labels in some of the panels in Figures 3, 5, 6, 7, 9, 10 (top row), 12 (top row), 13 (top row), S4, S5, S6, S7, S10, and S12. Several figures don't include a color bar (e.g., Figure S5-7). Please carefully revise all figures. Note that this point was already raised in the previous round of reviews, but it was not addressed.

3) Figure 3D – Loss aversion over time: Instead of running a separate between group comparisons for each time-point, it would be more appropriate to run a single two-way ANOVA with within-subject factor time and between-subject factor group. A significant group-by-time interaction would support the conclusion that loss aversion diverges between the two groups across time.

4) Why was the lateral OFC (area 47/12) not included here, given that work by Suzuki et al. 2017 suggests that lateral OFC represents attribute-specific values?

5) It is not fully clear what is plotted in Figure 5-7. Are these the averages across all RDM cells with a certain δ G/L/EV in Figures S5-S7? Does this include the δ G/L/EV = 0? How is neural coding strength defined in the lower rows? Tick labels would have helped.

6) Figure 9 shows that subject-specific ANNs correlate with subject-specific RDMs. To claim that these models capture individual patterns of OFC activity, it would be important to show that these correlations exceed those with group-level ANNs. Moreover, to claim specificity for plastic ANNs, it would be necessary to show superiority of predictions from plastic vs static ANNs.

[Editors' note: further revisions were suggested prior to acceptance, as described below.]

Thank you for resubmitting your work entitled "Efficient value synthesis in the orbitofrontal cortex explains how loss aversion adapts to the ranges of gain and loss prospects" for further consideration by *eLife*. Your revised article has been evaluated by Michael Frank (Senior Editor) and a Reviewing Editor.

The manuscript has been improved but there are some remaining issues that need to be addressed, as outlined below.

*Reviewer #1 (Recommendations for the authors):*

The authors have addressed my comments. I think the manuscript makes an interesting contribution to the field.

*Reviewer #2 (Recommendations for the authors):*

Overall the proposed model presents an interesting possible explanation for types of context-dependent loss aversion. The manuscript has improved over the course of revision and will be a worthwhile contribution to the literature. I have a few remaining comments that would help improve the clarity and accessibility of the results if they can be addressed before publication, but these are relatively minor.

1) While the figures have been substantially improved, several are still missing a description of the colors in the figure or legend, and instead have the placeholder phrase "(color code)" in the figure legend.

2) I appreciate your response to my previous comment about "undoing" adaptation (R2 Comment 3), but it is not clear to me in your response whether you are describing the computational role of "offer value" units in your model specifically, or just giving a hypothetical scenario. If I understand right, your model produces choice via a comparison of Vt for two options, and the"offer value" units are part of the integration layer (i.e. an input to Vt rather than the signal being compared directly). Is the idea that this would lead to stable preferences even without "undoing" adaptation downstream? Or would your model predict that preferences do shift in responses to "offer value" adaptation, and you suspect that past studies may not have been able to see it? Or are you just trying to say that there are several hypothetical possibilities, and in the specific task you are modeling it is not necessary to modify the weights? (As an aside, I also disagree with the argument that Rustichini et al. are interpreting a null result as evidence of absence – they start by predicting how preferences would change if choices arose from a simple comparison of offer value firing rates, then show that actual choice behavior does not match this prediction.)

3) In line 456 you discuss the spatial specificity of results, but unless I'm missing something this doesn't involve a direct comparison between regions. It may be worth reducing this claim.

*Reviewer #3 (Recommendations for the authors):*

Thank you for a responsive revision, I have no further points other than that the paper would still benefit from careful spell-checking.

---

## [Author Response]

Essential revisions:1) As you can see in the individual comments, all reviewers thought that your paper addresses an important topic. However, a central concern raised by all reviewers was that a substantial part of the model performance is driven by the activation function – yet, throughout the manuscript, you mainly discuss the role of Hebbian plasticity and largely ignore the effects of the activation function. There was agreement among reviewers that this is not warranted and that your manuscript requires substantial reframing, clarification, justification, and discussion. Reviewers would expect an adequately revised manuscript to look quite different from the current version, with an equal focus on all factors that allow the model to account for the observed changes across different ranges.2) There are additional comments in the individual critiques that would be important to address, specifically regarding aspects of the analysis and interpretation.

First of all, we would like to thank you for your insightful and constructive criticism. Some of your comments helped us understand that our previous interpretations were unsatisfactory, and we eventually changed our mind on many important aspects of our work. In particular, extensive posthoc numerical simulations made us realize that Hebbian plasticity, as described in the previous version of the manuscript, *does not* induce value range adaptation in most contexts. This meant that our previous results were somehow illusory, in the sense that they would not generalize to other types of datasets. We thus started from scratch, and derived another -novel- computational framework, which is grounded on formal models of efficient coding. We will detail this below. The resulting manuscript is entirely different in its content, and we hope to have significantly improved the quality of our contribution.

Second, we would like to apologize for the time it took us to revise this manuscript. The reason for this delay is twofold. First, the first author (Jules Brochard) first moved to a new lab for his postdoc, and then eventually quit academia. This implied that we could only progress very slowly. Second, we made so many changes – on both computational modelling and data analysis sides – that this revision really required a lot of time to complete. Nevertheless, we hope that the resulting work will satisfy most of your concerns. We reiterate that this would not have been possible without you making us think again about our work.

Before we respond to your comments point by point below, we would like to summarize our revisions. We believe this is necessary to clarify some of our responses.

As we mentioned above, raw Hebbian plasticity turns out not to be a good model for range adaptation. We realized this when attempting to address one of your questions, about *why* Hebbian plasticity would induce range adaptation. We first tried to derive analytical results regarding the impact of Hebbian plasticity: this did not work. We then resorted to numerical simulations on a wide range of conditions, but this eventually demonstrated that Hebbian plasticity does not yield value range adaptation. More precisely, it only does so in very specific settings of the ANN internal connectivity, and these settings cannot be summarized in a simple manner and/or justified from first principles. So what does this imply for our previous analyses? When fitting the plastic ANNs to empirical data, we unknowingly identified idiosyncratic variants of these settings, eventually extracting “Hebbian explanations” for range adaptation. But we now know that these explanations were, at best, anecdotal: they would not generalize.

This was a rather disappointing realization. At this point, we wondered how to move forward. We thus reversed the logic of our reasoning and asked: what sort of change in the ANN structure would eventually yield range adaptation? We were looking for a computational principle that would suggest *why* neurons would range-adapt, i.e. why this would be adapted. We took inspiration from the theoretical literature on efficient coding, which highlighted that range adaptation is the mechanism by which neurons minimize the information loss that is induced by their limited firing range. However, existing efficient coding models had been derived for perceptual brain systems, where neurons *transmit* the information that they receive. Put simply, the idea here is that a neuron’s input is the physical quantity that is signaled to the brain (e.g., light intensity within a certain frequency band), whereas the neuron’s output is the percept (e.g., perceived amount of red). In turn, range adaptation (to a neuron’s input signal) directly induces perceptual context-dependent effects. This is not case in our context: value is the outcome of an integration mechanism, over multiple (and possibly conflicting) decision-relevant information. We thus extended existing efficient coding models to ANNs that operate such *value synthesis.* More precisely, we show that a simple form of self-organized plasticity between the ANN’s attribute-specific and attribute-integration layers does mitigate the information loss induced by the limited firing range of neural units. In what follows, we refer to this as *efficient value synthesis*.

As you will see, this type of self-organized plasticity shares with Hebbian plasticity its simple form, i.e. connections progressively change as a function of the output responses of pairwise connected units (see Equation 2 in the revised manuscript, as well as its mathematical derivation in the revised Methods section). However, it is not, strictly speaking, Hebbian (i.e. it does not reinforce connections that yield co-activation of source and target units). If anything, it is similar to -though simpler than- ANN training rules underlying infomax variants of independent component analysis or ICA, which are essentially anti-Hebbian (Bell & Sejnowski, 1995; Nadal, 1994). Importantly, and for the same reason than in ICA network models, it turns out that this type of self-organized plasticity is incompatible with Gaussian activation functions. When revising this work, we thus focused on sigmoidal activation functions.

In the revised manuscript, we first summarize the mathematical derivation of efficient value synthesis and highlight its neural and behavioral consequences (in terms of the sensitivity to decision-relevant attributes, i.e. here: prospective gains and losses). Importantly, we show that efficient value synthesis induces value range adaptation in a wide range of contexts. We then test these neural and behavioral predictions using model-free data analyses in five subregions of the OFC. Finally, we reperform our previous model-based analyses using ANNs with and without self-organized plasticity, and show that only the former do yield accurate out-of-sample predictions of neural and behavioral data. Note that we also compare this model to a simpler range adaptation model, which operates at the level of attributes.

These changes eventually translated in significant modifications in all sections of the manuscript (Intro, Results, Methods, Discussion and Supplementary Materials). In fact, we had to replace most of the content of the previous version of the manuscript. This implies that some of the raised reviewers’ comments and questions may not be relevant anymore.

Nevertheless, we tried to address each one of them below, while referring to the novel computational framework that propose here. In any case, we believe our work has been significantly strengthened, and we thank once again the reviewers for their insightful comments and constructive criticism.

Reviewer #1 (Recommendations for the authors):1) Please explain on page 4 of the introduction why Hebbian plasticity should lead to spill-over effects. This is not very intuitive.

This is a fair point. In brief, numerical simulations show that neither does Hebbian plasticity induce value range adaptation, nor does it lead to spillover effects. This is not the case for efficient value synthesis. However, we have toned down the importance of spillover effects. This is because we don’t think they are a reliable signature of range adaptation in integration units. We understood this by deriving an efficient coding model that at the level of attributes: this model also predicts spillover cross-attribute effects. More precisely, what discriminates the two models is the direction of the within-attribute effects, whereas the cross-attribute spillover effects are similar.

Note that the mathematical derivation of the model of efficient coding of attributes, the summary of its neural and behavioral predictions, as well as the results of its related model- based data analyses are reported in the revised Supplementary Materials.

2) Interpretation of the null findings in the fMRI data (page 4 of results) is problematic because it is unclear whether they reflect a true null effect or a lack of sensitivity. Although this is true for all null results, it is particularly problematic for re-analyses, as the study was not designed or powered to test this question. It would be best to remove these results from the paper.

You are referring here to the null findings of univariate fMRI data analyses. We have now moved these results to the Supplementary Materials.

3) There is a fundamental difference between gaussian and sigmoidal activation functions. It would be important to include an adequate discussion of the assumptions and implications of these different functions in the main text.

Sigmoidal activation functions are a simple summary of the known physiological neural response to an electrical input: the firing rate of neurons is known to be both lower and upper bounded. In our previous manuscript, we also considered Gaussian activation functions. In brief, they are a phenomenological modelling assumption that captures neural receptive fields that span a bounded subregion of stimulus’ properties (e.g. spatial field of view in V1 neurons). We note that Gaussian activation function can be understood as the physiological output of a neuron that is reciprocally coupled with an inhibitory unit (where both have sigmoidal activation functions).

In any case, we do not consider Gaussian activation functions anymore in the revised manuscript. This is because, under Gaussian activation functions, the self-organized plasticity mechanism that mitigate information loss yields unstable dynamics. We comment on this point in the revised Discussion section (lines 536-561).

4) The out-of-range predictions of the best model (Figure 3, lower-right panel, HP-ANN (gauss)) are not very convincing when considering the entire EV range. Model performance should be compared for the entire EV range, not just the common range. What does this mean for the proposed mechanism?

In the previous version of the manuscript, out-of-sample predictions of ANNs equipped with Hebbian plasticity were indeed inaccurate outside the common EV range. In particular they exhibited a non-monotonic relationship between EV and gambling rate. This is not the case anymore when relying on ANNs endowed with self-organized plasticity that operates efficient value synthesis under sigmoidal activation functions (see Figure 7 in the revised manuscript). Of course, the ensuing behavioral out-of-sample predictions are still not perfect. More precisely, they tend to slightly under-predict the observed context-dependency effect of peoples’ risk attitude. Nevertheless, the accuracy of our out-of-sample predictions has clearly improved. In addition, we now provide complementary analytical results and model-free evidence (see below) that strengthen our computational claims.

5) The paper focuses on Hebbian plasticity as a mechanism for context effects but judging from Figure 3, the choice of activation function has a comparable effect. Indeed, HP does almost nothing for models with sigmoidal activation functions, and HP only improves the out-of-range prediction for the common EV range but does very little for the uncommon range. A more balanced presentation that also discusses the type of activation function as a mechanism of adaptation would be important.

You are right: in our previous version of the manuscript, Hebbian plasticity could only capture temporal range adaptation in peoples’ risk attitude when combined with Gaussian activation functions. Let us reiterate that, given that Hebbian plasticity only yields range adaptation under specific ANN connectivity patterns, this should be considered as anecdotal evidence for this mechanism. Since we do not consider Gaussian activation functions anymore, this comment is now irrelevant. Nevertheless, for the sake of completeness, we would like to clarify our previous analyses.

In brief, neither gaussian nor sigmoidal activation functions can induce temporal range adaptation effects by themselves: those can only be the outcome of dynamical mechanisms that change the network’s response over time (as is the case for efficient coding of attributes or efficient value synthesis). This is why we did not consider the form of the units’ activation functions as a potential cause for temporal range adaptation. If anything, it should be considered a potential moderator of the impact of dynamical mechanisms such as plastic changes in the ANN’s internal connectivity.

6) Are the RSA results in Figure 4 based on the ANNs with gaussian activation function? It would be important to show RSA results for all 4 ANNs in Figure 4, so it is possible to compare the results across models. Also, please add labels to the plot axes.

Again, we have abandoned Gaussian activation functions in the revised manuscript. Nevertheless, we do now report the results of both out-of-sample behavioral and neural predictions for all models. For RSA, the results of the efficient value synthesis (resp., efficient coding of attributes) scenario are summarized in Figure 8 of the main text (resp., Figure S12 of the Supplementary Materials).

We note that we have extended our previous RSA analyses. In addition to the quantification of trial-by-trial similarity between ANNs’ integration layer and fMRI activity patterns, we now also measure their similarity on gain-, loss- and EV-dependent RDMs (the latter are derived by binning trials according to either gain, loss or EV). We did this because we could obtain quantitative predictions regarding the related information content within the integration layer of ANNs operating efficient value synthesis (or efficient coding of attributes). This eventually provides more opportunities for evaluating the evidence strength for or against our candidate computational models. In particular, this allows us to perform model-free data analyses, which are designed to test between-group differences in the neural encoding strength of gains, losses and EVs.

7) The proportion of offer and chosen value neurons shown in Figure 6 is opposite to what has been reported in the OFC of non-human primates. It would be good to discuss this discrepancy. Also, are the same proportions found for all 4 ANNs?

This comment refers to our previous posthoc analysis of ANN integration units’, which showed that ANNs equipped with Hebbian plasticity reproduced the known diversity of coding properties in OFC neurons. In the revised version of our manuscript, we have reperformed this analysis (this time with ANNs operating efficient value synthesis), and it yields qualitatively similar results. Nevertheless, we have decided to tone down this point, and have moved these results to the Supplementary Materials.

8) Figure 4: the spheres shown for BA11 are in the posterior medial rather than the lateral OFC. Please double-check the anatomical location of these ROIs. Are they really in the lateral OFC? Also, it would be good to provide center coordinates for the ROIs. In general, it would be important to better describe how the ROIs were generated. What were the search terms used in NeuroQuery? Also, NeuroQuery generates meta-analytic activation maps, not maps of anatomical structures. It would be better to use actual anatomical ROIs for fMRI data analysis.

We agree that our previous ROI partition of the OFC was somehow arbitrary. In particular, although the functional definition of the vmPFC is established in the fMRI community, its anatomical definition remains vague. We have now re-performed the RSA analyses based upon standard Broadman areas of the OFC, in particular: BA11 (splat into its lateral and medial parts), BA13, BA14 and BA32. This parcellation is based upon masks obtained from the BRAINNETOME atlas (https://atlas.brainnetome.org/): it tiles the entire OFC, except its most lateral part (which is BA12). We now describe in full details how the ROIs were obtained from BRAINNETOME anatomical masks. We also provide the ROI barycenter coordinates in a Table (see revised Methods section).

Reviewer #2 (Recommendations for the authors):1) Range adaptation is not shown directly in ANN units. Specific questions:

We now include detailed analyses of ANN numerical simulations with and without self- organized plasticity. We describe these analyses in the Methods and Results section of the revised manuscript. In brief, they allow us to address all the concerns below, and more. We now summarize the results of these analyses when answering your comments:

a) How do ANN units respond across different input values? How variable are these response patterns across units?

As a reminder of the overall structure of the ANN, the first layer is divided into two sublayers, which receive prospective gains (resp., prospective losses). Units of the first layer (so-called “attribute units”) send their outputs to units of a second layer (so-called “integration units”).

Subjective value is readout from the output activity of this second layer.

In brief, attribute sublayers form a population code of their respective inputs, i.e. their units are selective of either prospective gains or prospective losses (depending on the specific sublayer), which they belong to. In contrast, integration units exhibit mixed selectivity, with heterogeneous response profiles across units. We refer the reviewer to Figures 10 and 12 of the revised manuscript for representative examples of integration units’ receptive fields (over the gain/loss domain).

b) How do these response patterns vary with range?

Under the efficient value synthesis scenario, attribute units do not change their response patterns with the range of either gains or losses. However, self-organized plasticity eventually modifies the receptive field of integration units. This modification tends to blur the tiling of the domain that is spanned by prospective gains and losses inputs. More importantly, one can show that this eventually modifies the information content within the ANN’s integration layer in a systematic manner. In particular, the encoding strength of prospective gains (resp., losses) decreases when the spanned range of gains (resp., losses) increases.

c) Do these patterns (at the individual unit or mean level) resemble neuronal data from the orbitofrontal cortex (OFC)? Should it be expected to?

Qualitatively speaking, integration units do exhibit response profiles that are reminiscent of typical OFC neurons electrophysiological activity during value-based decision making. For example, they reproduce the diversity of coding that has been repeatedly observed in OFC neurons during value-based decision making (cf., “offer value cells”, “chosen value cells”, and “choice cells”, see Figure S8 of the revised Supplementary Materials). More importantly, integration units also exhibit the known properties of value range adaptation in these same neurons (see Figure 2 in the revised Results section).

Now, whether this sort of ANN model produces “realistic” electrophysiological activity profiles beyond this kind of statistical relationship is questionable. The reason is twofold. First, they are agnostic w.r.t. within-trial temporal dynamics. Second, there is some level of arbitrariness in the modelling assumptions (cf., e.g., structural constraints) that cannot be finessed using either behavioral or neuroimaging data. What we argue is robust in these ANN models is the information content that they carry, which is distributed over the activity profiles of their artificial neural unit layers. This is the main reason why we resort to variants of RSA for comparing their predictions to multivariate fMRI activity patterns.

We now comment on these issues in the revised Discussion section (lines 521-535).

d) The example units that are shown (Figure 7) seem potentially different from previously reported neuronal data from OFC. For example, the dynamic range of the model covers only a narrow subset of the input space and varies substantially across conditions, whereas firing rates in OFC neurons tend to span the full range of possible values, and firing rates change only slightly between conditions [ref,ref]. Is this discrepancy just a side effect of showing results in terms of loss units rather than expected value? How should we interpret this apparent discrepancy?

This point is directly related to the above comment Re: whether one should expect the ANN units to resemble neuronal data from OFC. We note that Figure 7 (in the previous version of the manuscript) was meant as a schematic depiction of the Hebbian plasticity mechanism under a strawman “population code” scenario. In our revised manuscript, we have now replaced it with graphical summaries of numerical simulations of ANNs. As you will see, representative ANN units exhibit receptive fields that have a rather arbitrary shape (i.e., with mixed gain/loss selectivity ; see Figures 10 and 12 of the revised manuscript). However, the induced statistical relationship between integration units’ output response and value is much simpler (in particular, it tends to be rather monotonic). In addition, this relationship typically spans the full range of values, with firing rates that show small changes across conditions (cf. Figure 2).

e) ANN unit responses are compared to neuron classes observed in OFC (Figure 6). What does the mean ANN unit in each category look like, and how does this compare to the OFC responses referenced?

In our context, “choice units” show response outputs that vary in a quasi-categorical manner with value (i.e. that discriminate trials in which the value of gambling is either positive or negative), “chosen value units” exhibit a ReLU-like relationship with value (null when the value of gambling is negative, and increasing with value when it is positive), and “offer value units” show a quasi-linear relationship with value.

2) Modeling results focus on Hebbian networks with gaussian activation functions. Specific questions:a) Is there a physiological motivation for the model, or is this primarily for mathematical convenience?

In brief, the Gaussian activation functions that we used in the previous version of this manuscript were a mathematical convenience. Having said this, a Gaussian activation function can be understood as the physiological output of an excitatory unit that is reciprocally coupled with an inhibitory unit (where both have sigmoidal activation functions).

But we have abandoned this type of activation function anyway…

b) Do the main qualitative results (range effect on risk sensitivity) require a non-monotonic activation function?

We now provide extensive theoretical analyses of adaptation effects induced by self- organized plasticity within the network. In principle, these results would generalize to any monotonic activation function. Importantly, this would not hold for non-monotonic activation functions, because those would induce unstable plasticity dynamics. This is the reason why we now only focus on sigmoidal activation functions.

c) How do the properties of the response functions (saturation and (non)-monotonicity) affect responses of ANN units?

Intuitively, units with sigmoidal activation functions will be generally more active than units with Gaussian activation functions. This is because their effective receptive field is not upper-bounded. For the same reason, they will exhibit less functional specificity than units with Gaussian activation functions. But again, this is now irrelevant.

d) There are visible differences between model behavior for Hebbian networks with sigmoidal and gaussian activation functions. How should these differences be interpreted? Does this lead to any predictions or constraints on what physiological implementation is consistent with this algorithm?

Let us first respond in the frame of our previous computational model and analyses. In brief, we had a priori expected that both types of units, when equipped with Hebbian plasticity, would exhibit range adaptation effects. We had thus hoped to show that the ability of the Hebbian mechanism to yield accurate out-of-sample choice predictions would not depend upon the type of underlying activation functions. This turned out not to be the case. Note that we did not believe that there was a physiological lesson to be learned here. Rather, we thought that this difference may be an unavoidable artefact of our data analysis procedure.

Recall that fitted ANN models can generalize across groups (or contexts) if and only if they neither underfit nor overfit the choice data (and hence accurately discriminate temporal changes in gambling value that are due to adaptation from unsystematic choice variations). Although we had no a priori reason to expect that ANNs with sigmoidal activation functions would be more prone to underfitting or overfitting, it is reasonable to think that the tendency to underfit/overfit choice data may depend upon the type of units’ activation function.

Now, although we do not use Gaussian activation functions in the revised manuscript, this point still deserves a further comment. As we highlighted before, efficient value synthesis yields unstable plasticity dynamics under non-monotonic (e.g. Gaussian) activation functions. To understand this, recall that the self-organized plasticity rule derives from aligning the connectivity change with the gradient of information loss w.r.t. connection strengths. This gradient explodes when inputs fall within domains where the derivative of the activation function approaches zero. This unavoidably happens with non-monotonic activation functions because the plasticity mechanism eventually focuses the weighted inputs within the vicinity of their mode. In other terms, one may argue that only monotonic activation functions are compatible with the efficient value synthesis scenario. We note that this kind of issue was already highlighted in computational studies of network models of ICA (Bell & Sejnowski, 1995; Nadal, 1994). We now comment on this in the revised Discussion section (cf. lines 536-551).

3) Out-of-sample predicted choices in the Hebbian model with gaussian activation function seem unintuitive for more extreme parts of the value range. E.g. for the out-of-sample wide range predictions, the model seems to over-predict risk aversion to the point that choice probabilities saturate and ~0.6 for increasingly high-value options. For the narrow range, out-of-sample prediction behavior even appears to be non-monotonic, leading to increased acceptance of extremely high loss options.a) How should this be interpreted?

This was an artefact of our previous data analysis procedure. Recall that ANNs effectively are a nonlinear mapping between their inputs (here: prospective gains and losses) and their outputs (here: gambling choices). This mapping was made of a mixture of basis functions, whose receptive fields typically tile the gain/loss domain over which the ANN is trained. This means that, without additional constraints, this mapping poorly generalizes outside the range of gain and loss inputs with which it was trained on. Having said this, our modified model shows much better-behaved out-of-sample predictions…

b) Does this depend on model parameters like update rate or covariance threshold?

No, this artefact did not depend upon model parameters.

c) In these parts of the range, how does the Hebbian model compare to alternate models, such as the other ANNs or logistic regression?

In this part of the range, logistic regression or ANNs with sigmoidal activation functions did generally better than ANNs with Gaussian activation functions. But this is not the case with the efficient value scenario (which relies upon sigmoidal activation functions).

4) The relationship between neural responses and ANN activity relies on representational similarity analysis (RSA). However, significant RDM correlations could arise as a byproduct of the fact that both ANN output and BOLD activity in select regions correlate with behavioral choice patterns. Is there evidence that the correlation between Hebbian ANNs and BOLD activation reflects more than average choice patterns?

This is a fair point: you are asking whether the similarity between the model and neural data may not simply be driven by the similarity between the model and choice data (e.g., a model that would not fit choice data at all would be very unlikely to be similar to neural data). We do agree that this is in principle possible here. For example, if, for some reason, static ANNs underfit choice data (at least when compared to plastic ANNs), then they might have a better chance to look like trial-by-trial fMRI signals in the OFC. A first hint here comes from Figure S3 in our revised manuscript: one can see that plastic and static ANNs yield very similar choice postdiction error rates. Therefore, they are unlikely to differ in terms of their behavioral explanatory power.

But our model-free fMRI data analyses provide, in our opinion, a stronger counter argument here. In brief, only ANNs that operate efficient value synthesis do predict the observed change in the information content induced by the difference in spanned gain range. In particular, the information content about prospective gains and losses (as opposed to integrated value) is conditionally independent from choice data. Note that ANNs that operate efficient coding of attributes do not make accurate neural predictions (cf. Figures S10 and S12 in the Supplementary Marterials).

5) The authors make the strong claim that this model is the mechanistic explanation for adaptation in orbitofrontal cortex, but there is little comparison with previous models. Divisive normalization and other forms of adaptation to the value range are discarded based on a qualitative argument from the behavioral data. However, given that the Hebbian ANNs also produce some counterintuitive behavioral predictions, it is not obvious that they are better at accounting for observed patterns in neuronal adaptation or behavior. Addressing the following questions could clarify whether there is an argument for Hebbian ANNs over alternate mechanisms of adaptation:a. How do behavioral predictions and RSA results from HP-ANNs compare quantitatively to results from other models of adaptation, including models of adaptation at the input stage?

This is an important point. In the revised version of the manuscript, we now consider a model of adaptation at the level of attributes. Its mathematical derivation, its neural and behavioral predictions, as well as the ensuing model-based data analyses are reported in the revised Supplementary Materials (cf. “value synthesis under efficient coding of gains and losses”). At the behavioral level, it makes predictions that are similar to the efficient value synthesis scenario. At the neural level however, it – somehow surprisingly- exhibits distinct properties. In particular, it *does not* induce value range adaptation in integration units (see Figure S9).

Moreover, it predicts that the neural encoding strength of gains (resp., losses) *increases* when the range of spanned gains (resp., losses) increases. This clearly contradicts our model-free analyses of fMRI data in the OFC.

Regarding divisive normalization, we do not see how it would provide a simple alternative explanation for this dataset. We acknowledge that divisive normalization provides an elegant explanation for specific forms of instantaneous context-dependency effects. For example, when deciding between three options, divisive normalization would predict irrational decoy effects (Louie et al., 2013; Steverson et al., 2019). But these kinds of models are not standard explanations for temporal range adaptation effects. To our knowledge, there exists only one variant of divisive normalization models that aims at capturing temporal range adaptation (Zimmermann et al., 2018). The model is very complex and relies on coupling slow and fast winner-take-all networks. Critically, it treats option values as inputs, and does not consider situations where subjective value has to be constructed from the integration of multiple decision attributes. We believe that it is beyond the scope of this work to extend this kind of model to networks that operate value synthesis.

b) Can Hebbian ANNs account for other previously observed patterns of behavior across different value ranges, such as stability of relative values across ranges in two-option choices and range-dependent decoy effects []?

Addressing this sort of issues would require extending the model to situations in which two (or three) option values are simultaneously represented and/or compared by the network. These additional mechanisms are beyond the scope of the current work (but we will be addressing these points in subsequent publications).

c) Are there specific predictions that arise from Hebbian networks that could be tested in later work and used to differentiate between competing models?

One possibility here is to target, using invasive neurophysiological approaches, the mechanisms that underlie plasticity (e.g., LTP and LTD). For example, specifically suppressing LTP and/or LTD (in value-coding OFC neurons) should prevent or distort value range adaptation. One could then compare choice behavior immediately after exposing subjects to (high or low) ranges of gains and/or losses, with and without LTP/LTD suppression.

Reviewer #3 (Recommendations for the authors):1) The paper assumes loss aversion as the primary behavioral factor, which is fine. However, it may be worth briefly mentioning the limitation that with the present experimental design it is impossible to dissociate loss aversion from risk aversion (see e.g. Williams et al., 2021, PNAS).

This is a fair point and we have included a comment on this in the revised Discussion section.

[Editors' note: further revisions were suggested prior to acceptance, as described below.]

All reviewers agreed that the revised manuscript has been improved. However, given the revised manuscript is essentially a fundamentally new manuscript, there is a new set of comments that would need to be addressed, as outlined below.

First of all, we thank the Editor and reviewers for providing us with another opportunity to improve our work. We have tried to address each comment (see our response below). In particular, we included a few additional results reports, modified most figures, moved some material from the Methods section into the Supplementary Material, and included novel discussion points in the Discussion section. We hope that you will agree with these changes.

Reviewer #1 (Recommendations for the authors):I appreciate the authors' effort and dedication in re-working the manuscript. The revised manuscript is a fundamentally new and improved paper with new conclusions. I believe it could make an important contribution to the field and I remain enthusiastic. However, given most of the manuscript has changed, I have new comments that I think should be addressed

Thank you very much for your positive appreciation of our work.

1) In general, the manuscript is quite long with extensive supplementary analyses. I believe the paper could be streamlined by highlighting the most important aspects and reducing the extent to which details are discussed in the main text.

We understand your comment. However, we found it difficult to remove material in the main text without harming the readability of the manuscript. In the hope of conciliating these two imperatives, we have moved some of the content of the Methods section in the Supplementary Materials (see also our response to point #12 of reviewer #2). We hope that you will find our revised manuscript concise enough.

2) All relevant information to understand the plots should be embedded within the figure rather than just the figure legends. It is cumbersome for readers to constantly have to consult the legends to understand what is shown in the figure. For instance, there is no label for the different colored plots (red/back, red/blue) or line styles in ANY of the figures. Also, some legends (Figure 4) refer to a color code, but this code is not provided. Moreover, there are no axis ticks and/or axis tick labels in some of the panels in Figures 3, 5, 6, 7, 9, 10 (top row), 12 (top row), 13 (top row), S4, S5, S6, S7, S10, and S12. Several figures don't include a color bar (e.g., Figure S5-7). Please carefully revise all figures. Note that this point was already raised in the previous round of reviews, but it was not addressed.

We have gone through all the manuscript’s Figures and modified them to improve readability as much as possible. In particular, we have now inserted self-contained axis ticks, colorbars and legends whenever they were missing. Note that most Figures number have changed, because we had to tie Figures in the supplementary material to Figures in the main text.

3) Figure 3D – Loss aversion over time: Instead of running a separate between group comparisons for each time-point, it would be more appropriate to run a single two-way ANOVA with within-subject factor time and between-subject factor group. A significant group-by-time interaction would support the conclusion that loss aversion diverges between the two groups across time.

This is a fair suggestion. We tried this but found no significant time-by-group interaction (p=0.43, F=0.61, R2=0.6%), which is why we report separate group comparisons. We now also report this null finding prior to reporting separate group comparisons.

4) Why was the lateral OFC (area 47/12) not included here, given that work by Suzuki et al. 2017 suggests that lateral OFC represents attribute-specific values?

In brief, we had no particular reason for not including area 47/12. We simply considered OFC subregions that were close neighbors to (the typical fMRI definition of) the ventromedial PFC. This effectively disqualified BA12/47, which is positioned on the most lateral part of the OFC. Having said this, we do not think that including BA47/12 is critical for our empirical demonstration. This is because fMRI activity patterns in BA11 already validate the model’s predictions (and we essentially rely on other OFC subregions as control ROIs). Given that inserting this additional ROI would mean adding more material to the main text and induce further delays in the revision of this work, we thus decided not to do it. We hope you understand and agree with us.

5) It is not fully clear what is plotted in Figure 5-7. Are these the averages across all RDM cells with a certain δ G/L/EV in Figures S5-S7? Does this include the δ G/L/EV = 0? How is neural coding strength defined in the lower rows? Tick labels would have helped.

Yes, upper panels in Figure 5 to 7 show the group mean of within-subject RDMs binned by differences in variables of interest (i.e.: G, L or EV). And yes, the group mean of G/L/EV-specific RDMs can be eyeballed in the Supplementary Material (Figure 5 —figure supplement 1, Figure 6 —figure supplement 1 and Figure 7 —figure supplement 1, respectively). In our previous manuscript, the y-axis limits were kept identical across OFC subregions, but this was not apparent because we had removed tick labels. We have included them all now.

6) Figure 9 shows that subject-specific ANNs correlate with subject-specific RDMs. To claim that these models capture individual patterns of OFC activity, it would be important to show that these correlations exceed those with group-level ANNs. Moreover, to claim specificity for plastic ANNs, it would be necessary to show superiority of predictions from plastic vs static ANNs.

These are fair comments. We tried both suggestions but failed to detect significant differences (both for within-subject versus group correlations and for plastic versus static ANN variants). We thus have modified our results report to tone down these claims. In brief, we simply use this analysis to afford evidence that ANNs that operate value synthesis (whether plastic or static) do yield reasonably realistic predictions regarding fMRI activity patterns within the OFC.

Note that, for the sake of completeness, we modified Figure 9 to enable readers to eyeball and compare the RSA results of both plastic and static models.

[Editors' note: further revisions were suggested prior to acceptance, as described below.]

Reviewer #2 (Recommendations for the authors):Overall the proposed model presents an interesting possible explanation for types of context-dependent loss aversion. The manuscript has improved over the course of revision and will be a worthwhile contribution to the literature. I have a few remaining comments that would help improve the clarity and accessibility of the results if they can be addressed before publication, but these are relatively minor.1) While the figures have been substantially improved, several are still missing a description of the colors in the figure or legend, and instead have the placeholder phrase "(color code)" in the figure legend.

We have now modified all relevant figure legends to provide an explicit description of the color code (e.g., for Figure 3: “[…] A: the neural encoding strength of gains (color code: from blue -minimal encoding strength- to yellow -maximal encoding strength-) is shown as a function of the spanned range of losses (x-axis, range increases from left to right) and gains (y-axis, range increases from bottom to top)). […]”.

2) I appreciate your response to my previous comment about "undoing" adaptation (R2 Comment 3), but it is not clear to me in your response whether you are describing the computational role of "offer value" units in your model specifically, or just giving a hypothetical scenario. If I understand right, your model produces choice via a comparison of Vt for two options, and the"offer value" units are part of the integration layer (i.e. an input to Vt rather than the signal being compared directly). Is the idea that this would lead to stable preferences even without "undoing" adaptation downstream? Or would your model predict that preferences do shift in responses to "offer value" adaptation, and you suspect that past studies may not have been able to see it? Or are you just trying to say that there are several hypothetical possibilities, and in the specific task you are modeling it is not necessary to modify the weights? (As an aside, I also disagree with the argument that Rustichini et al. are interpreting a null result as evidence of absence – they start by predicting how preferences would change if choices arose from a simple comparison of offer value firing rates, then show that actual choice behavior does not match this prediction.)

In brief, we were arguing (i) that Rustichini et al’s argument is statistically unsound, and (ii) that a variant of our model would actually predict no behavioral change despite apparent value range adaptation in “offer value cells”. We take the latter as a relevant counter-example for how puzzling the result was in the first place. Now, since we believe this point is the most important, we have dropped our former statistical criticism in the revised manuscript. In addition, we have modified this paragraph to clarify our reasoning as much as possible:

Intriguingly however, value range adaptation in “offer value cells” had been observed without any significant behavioral preference change. Under the assumption that preferences between offers derive from the direct comparison of output signals from “offer value cells”, this is surprising. To solve this puzzle, later theoretical work proposed that value range adaptation is somehow “undone” downstream value coding in the OFC (Padoa-Schioppa & Rustichini, 2014). In our context, this would suggest that readout weights (*w^(k)* in Equation 2) would compensate for value-related adaptation, effectively thwarting the behavioral consequences of self-organized plasticity between attribute and integration layers. However, this reasoning critically relies upon the assumed computational role of “offer value cells”. In fact, this puzzle may effectively dissolve under other scenarios of how “offer value cells” contribute to decision making. Recall that this null result was obtained in a decision context where choice options were characterized in terms of the type of offer (i.e. juices that differ w.r.t. palatability), whose quantity was systematically varied. Here, value synthesis would effectively aggregate two attributes, namely palatability and quantity. Under this view, “offer value cells” simply are integration units that show a certain form of mixed selectivity, whereby units’ sensitivity to quantity strongly depends upon palatability. At this point, one needs to consider candidate scenarios of how the OFC may operate value synthesis for multiple options in a choice set. A possibility is that the OFC is automatically computing the value of the option that is currently under the attentional focus (Lebreton et al., 2009; Lopez-Persem et al., 2020), while storing the value of previously attended options within an orthogonal population code (Pessiglione & Daunizeau, 2021). In principle, this implies that the OFC is wired such that it can handle arbitrary switches in attentional locus without compromising the integration of option-specific attributes. In this scenario, integration units (including those that look like “offer value cells”) would adapt to the range of all incoming attribute signals, irrespective of which option in the choice set is currently attended. In turn, “offer value cells” would look like they are only partially adapting to the value range of a given offer type (Burke et al., 2016; Conen & Padoa-Schioppa, 2019). More importantly, to the extent that between-attribute spillover effects are negligible, changes in the range of offer quantities would distort the readout value profile along the quantity dimension without altering the palatability dimension. This would effectively leave the relative preference between offer types unchanged. Of course, this is only one candidate scenario among many. Nevertheless, we would still argue that the behavioral consequences of range adaptation in “offer value cells” actually depend upon their underlying computational role.

3) In line 456 you discuss the spatial specificity of results, but unless I'm missing something this doesn't involve a direct comparison between regions. It may be worth reducing this claim.

You are right. To make sure our claims regarding anatomical specificity are not over-interpreted, we have modified the relevant paragraph in the Discussion section as follows:

Although this clearly aligns with our model-free fMRI data analysis results, we do not claim that the evidence we provide regarding the anatomical location of value synthesis generalize beyond decision contexts that probe peoples’ loss aversion. In fact, our main claim is about whether and how efficient value synthesis operates within the OFC, as opposed to which specific subregion of the OFC drives the adaptation of loss aversion and/or related behavioral processes.